# Discovery of benzo[c]phenanthridine derivatives with potent activity against multidrug-resistant *Mycobacterium tuberculosis*

Yi Chu Liang,[1,2] Zhiqi Sun,[2] Chen Lu,[3,4] Andréanne Lupien,[5,6,7,8] Zhongliang Xu,[3,4] Stefania Berton,[2] Peng Xu,[1] Marcel A. Behr,[5,6,7,8] Weibo Yang,[3,4,9] Jim Sun[1,2]

**ABSTRACT** *Mycobacterium tuberculosis* (Mtb), the pathogen responsible for tuberculosis (TB), is the leading cause of bacterial disease-related death worldwide. Current antibiotic regimens for the treatment of TB remain dated and suffer from long treatment times as well as the development of drug resistance. As such, the search for novel chemical modalities that have selective or potent anti-Mtb properties remains an urgent priority, particularly against multidrug-resistant (MDR) Mtb strains. Herein, we design and synthesize 35 novel b̲enzo[c]p̲henanthridine d̲erivatives (BPDs). The two most potent compounds, BPD-6 and BPD-9, accumulated within the bacterial cell and exhibited strong inhibitory activity ($MIC_{90}$ ~2 to 10 µM) against multiple *Mycobacterium* strains while remaining inactive against a range of other Gram-negative and Gram-positive bacteria. BPD-6 and BPD-9 were also effective in reducing Mtb survival within infected macrophages, and BPD-9 reduced the burden of *Mycobacterium bovis* BCG in the lungs of infected mice. The two BPD compounds displayed comparable efficacy to rifampicin (RIF) against non-replicating Mtb (NR-Mtb). Importantly, BPD-6 and BPD-9 inhibited the growth of multiple MDR Mtb clinical isolates. Generation of BPD-9-resistant mutants identified the involvement of the Mmr efflux pump as an indirect resistance mechanism. The unique specificity of BPDs to *Mycobacterium* spp. and their efficacy against MDR Mtb isolates suggest a potential novel mechanism of action. The discovery of BPDs provides novel chemical scaffolds for anti-TB drug discovery.

**IMPORTANCE** The emergence of drug-resistant tuberculosis (TB) is a serious global health threat. There remains an urgent need to discover new antibiotics with unique mechanisms of action that are effective against drug-resistant *Mycobacterium tuberculosis* (Mtb). This study shows that novel semi-synthetic compounds can be derived from natural compounds to produce potent activity against Mtb. Importantly, the identified compounds have narrow spectrum activity against *Mycobacterium* species, including clinical multidrug-resistant (MDR) strains, are effective in infected macrophages and against non-replicating Mtb (NR-Mtb), and show anti-mycobacterial activity in mice. These new compounds provide promising chemical scaffolds to develop potent anti-Mtb drugs of the future.

**KEYWORDS** *Mycobacterium tuberculosis*, tuberculosis, non-replicating bacteria, multidrug-resistant TB, anti-mycobacterial compounds, benzophenanthridine, natural compounds

T uberculosis (TB) has existed throughout human history and is still threatening millions of lives each year (1). Since the discovery of *Mycobacterium tuberculosis* (Mtb) in the 19th century, the search for novel, specific, and effective antibiotics against Mtb has remained a major priority for TB drug discovery. Unlike Gram-positive or Gram-negative bacteria, the waxy, mycolic acid-rich cell wall of Mtb is the first barrier to overcome

Address correspondence to Jim Sun, jim.sun@ubc.ca, or Weibo Yang, yweibo@simm.ac.cn.

Yi Chu Liang, Zhiqi Sun, Chen Lu, and Andréanne Lupien contributed equally to this article. Author order was determined in order of increasing seniority working on the project.

The authors declare no conflict of interest.

See the funding table on p. 21.

in TB drug discovery. Small chemical molecules that are not permeable through the Mtb cell wall need to be coated or encapsulated to increase penetration *in vivo* or *ex vivo* (2, 3), necessitating extra research efforts. Additionally, under the pressure of host immunity and antibiotics, Mtb can adapt via a non-replicating state that facilitates non-heritable resistance or tolerance to most conventional antibiotics (4). As such, efficacy of TB drug candidates should be assessed using *in vitro* and *in vivo* models of non-replicating Mtb (NR-Mtb). A number of *in vitro* models have been established to generate low-metabolism Mtb cultures, including hypoxia, nutrient starvation, low pH, and the combination of multiple stresses (5). Of the frontline TB drugs, rifampicin (RIF) is the most well characterized to inhibit NR-Mtb under acidic conditions (6, 7). Another frontline TB drug, pyrazinamide, eliminates NR-Mtb *in vivo* and *in vitro* by targeting non-specific pathways (8). Furthermore, recently approved TB drugs, bedaquiline (9) and pretomanid (10) have demonstrated efficacy against NR-Mtb *in vivo* (6).

To find new mechanisms of action and new chemical scaffolds, natural products possess a range of benefits for drug discovery (3, 11). Natural products are ideal candidates for drug discovery due to their unique structural and scaffold diversity. Typically, natural products tend to be compounds with higher hydrophilicity and greater molecular rigidity than synthetic chemicals, which are favorable traits for drug development. Importantly, as most natural products are bioactive molecules refined by evolution to serve specific biological functions such as endogenous regulation and inter-organism competition, they are highly relevant to anti-cancer and anti-infection applications (12). Rifampicin is a successful example of a semi-synthetic drug derived from the natural compound rifamycin (13). Other well-known examples include streptomycin, the first successful antibiotic against TB that was isolated from *Streptomyces griseus*, as well as the anti-TB antibiotics kanamycin, produced by *Streptomyces kanamycetius*, and its semi-synthetic analogue amikacin (14).

Sanguinarine (SG), a natural benzo[c]phenanthridine alkaloid extracted from *Sanguinaria canadensis*, has been used as an antiseptic herbal essence and was used as an antimicrobial agent in toothpaste and mouthwash (15, 16). SG has shown antibacterial effects against various Gram-positive and Gram-negative bacteria (17–19). In a recent study, SG has been shown to be an inhibitor of the 2-ketogluconate pathway, which is important for glucose metabolism in *Pseudomonas aeruginosa* (19). There have also been reports demonstrating the synergistic effects of SG and antibiotics (17, 20). However, the antibacterial effects of SG against mycobacteria have not been explored and may represent a missed opportunity.

Herein, we designed and synthesized 35 unique derivatives of SG. Phenotypic activity screening identified five hits within this group of new compounds that possessed significantly improved anti-Mtb activity compared to SG. The two most potent compounds, benzo[c]phenanthridine derivative (BPD)-6 and BPD-9, possess low-micromolar inhibitory activity against multiple mycobacterial species but were inactive against other Gram-negative and Gram-positive bacteria, showing unique specificity against mycobacteria. Both compounds exhibit reduced cytotoxicity relative to SG and were active against intracellular Mtb. Importantly, BPD-9 was active against multiple multidrug-resistant (MDR) Mtb clinical isolates and showed *in vivo* anti-mycobacterial activity against *Mycobacterium bovis* BCG in infected mice. Generation of BPD-9-resistant Mtb mutants revealed an upregulation of the Mmr efflux pump (*rv3065*). Collectively, this study shows that novel benzo[c]phenanthridine derivatives have the potential to be developed into selective anti-mycobacterial drugs to provide new chemical options for TB antibiotic discovery.

## RESULTS AND DISCUSSION

### Generation of novel benzo[c]phenanthridine derivatives with antibacterial activity

Given that the antibacterial activity of SG has not been reported for *M. tuberculosis*, we first assessed whether SG possessed any inhibitory activity against Mtb mc$^2$6206, an auxotrophic strain that retains a similar drug susceptibility profile as its virulent parent strain H37Rv (21). Using the resazurin microtiter growth inhibition assay (REMA) (22), SG showed only modest activity against Mtb mc$^2$6206, reaching a minimum inhibition concentration (MIC$_{90}$) of 74 µM (Fig. 1A). To define and improve the activity of the pharmacophore responsible for the anti-Mtb activity of SG, we used a pharmacophore incorporation strategy to design new benzo[c]phenanthridine derivatives. Three points of pharmacomodulation were applied to modify the structure of SG: (i) derivatives modified at rings A and D of the benzo[c]phenanthridine moiety; (ii) simplification of

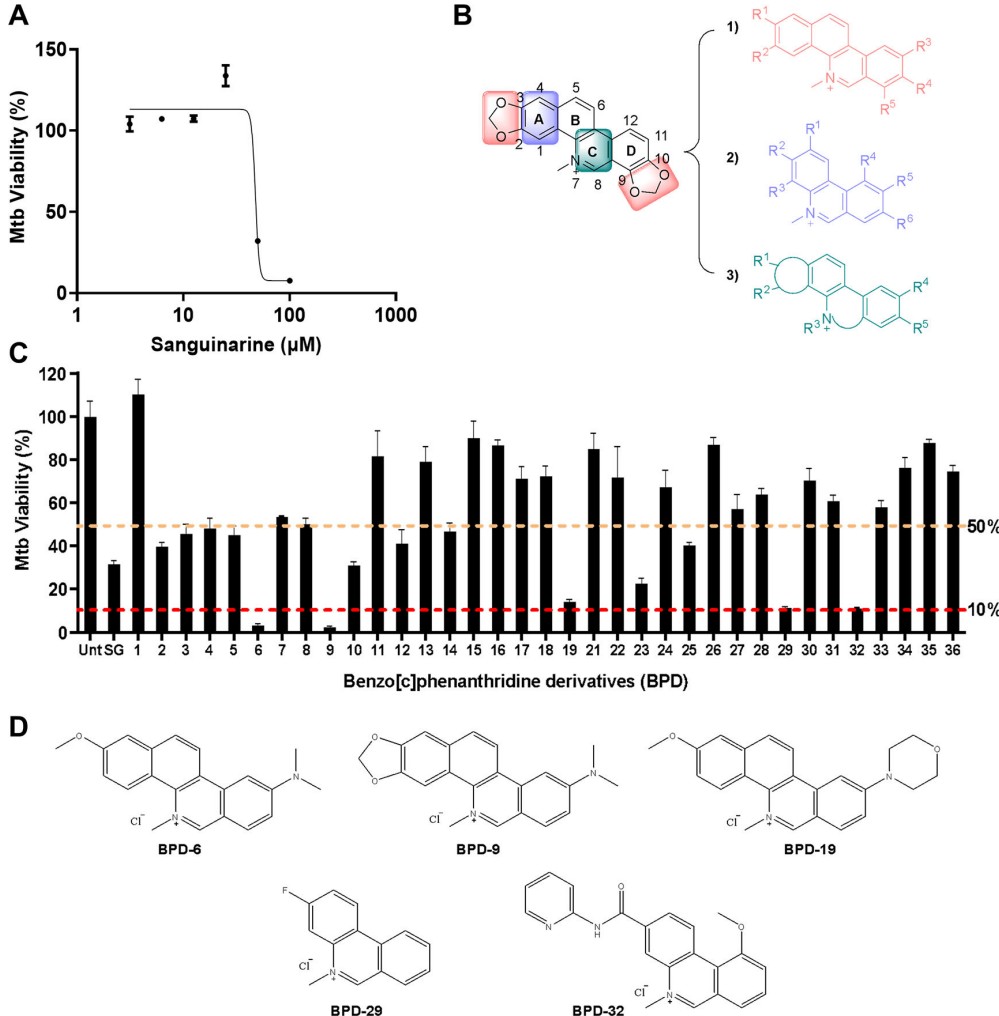

FIG 1 Generation of novel benzo[c]phenanthridine derivatives (BPDs) with antibacterial activity. (A) Activity of sanguinarine (SG) against *Mycobacterium tuberculosis* (Mtb) mc$^2$6206 was determined using the resazurin microtiter assay (REMA). Mtb viability is normalized to maximal bacterial growth in the absence of compounds as 100%. (B) Pharmacomodulation strategy to generate novel BPDs from SG. (C) BPD compounds were screened for activity against Mtb mc$^2$6206 at 40 µM using the REMA. Mtb viability is normalized to maximal bacterial growth in the absence of compounds as 100%. Unt, untreated. (D) Chemical structures of compounds with the highest Mtb inhibitory activity. Data in (A, C) represent the mean ± standard error of the mean (SEM) of three independent experiments.

the quaternary pyridinium skeleton of the phenanthridinium core; (iii) dearomatizing ring C and introducing diversified functional groups at C-8 or the N atom (Fig. 1B).

In total, 35 new benzo[c]phenanthridine derivatives (Fig. S1) were synthesized and screened for anti-Mtb activity using the REMA. In the initial screen using a fixed concentration of 40 µM, 15 (43%) of the new compounds were able to inhibit Mtb growth by at least 50% compared to untreated controls (Fig. 1C). Of these 15 hits, five compounds showed significantly improved activity compared to SG and inhibited the growth of Mtb by ≥85% (Fig. 1C and D). The two most potent hits, compounds #6 and #9, hereafter designated as BPD-6 and BPD-9 (benzo[c]phenanthridine derivatives) ( Fig. S2 and S3), showed greater than 90% inhibition and were, thus, selected for further characterization. Based on the structures of the active hits, the minimal pharmacophores of BPDs are rings B-C-D. The azanium of ring C is critical for anti-Mtb activity, and substitutions on rings B and D affect the activity of the compounds. The benzo-segment of ring B could be simplified to generate derivatives with considerable activity, and additional hydrogen receptors or donors could be combined for a productive gain in activity, such as in BPD-29 and BPD-32. The dioxolane on ring D is dispensable for activity, and introducing any substituents on C-9 or C-10 on ring D resulted in a loss of activity, which may be due to a blocking interaction with the target. However, introduction of nitrogen-containing groups at C-11 of ring D improved the potency significantly (BPD-6, -9, and -19).

## BPD-6 and BPD-9 significantly inhibit the growth of mycobacteria

The REMA growth inhibition assay was used to determine the dose-dependent inhibition activity of BPD-6, -9, -29, and -32 against Mtb mc²6206 (Fig. 2A). This assay showed that BPD-6 and BPD-9 possessed the most potent anti-Mtb activity, reaching an $MIC_{90}$ of 10 and 6 µM, respectively (Table 1). The activity of BPD-6 and BPD-9, thus, show an 8- and 14-fold improvement over SG, respectively, confirming that the new BPDs possess significantly increased anti-Mtb activity. Although SG is broadly active against several bacterial species, we wondered if this property was retained in our new compounds. To assess the specificity of BPD-6 and -9, we performed REMA growth inhibition assays on various mycobacterial strains. To our surprise, BPD-6 and -9 retained potent inhibitory effect against *M. bovis* BCG, *Mycobacterium kansasii*, and *Mycobacterium smegmatis* while showing no activity up to the maximum tested concentration of 40 µM against other Gram-positive and Gram-negative bacteria (Table 1). The specificity of an antibiotic to a particular pathogenic genus or species may have major advantages for treatment potential as it is less likely to adversely affect the host microbiome (23–25). The specificity of BPD-6 and -9 suggests a possible mechanism or target of inhibition that exists specifically in mycobacteria. It has been reported that SG binds to FtsZ in various bacterial species, including *Escherichia coli* (26), to inhibit their growth by preventing cell division (18). Although FtsZ is highly conserved at the amino acid level in prokaryotes, Mtb FtsZ has higher homology within *Mycobacteria* species (>90% similarity and identity) compared to bacterial species (those in Table 1) from other genera (~65% to 70% similar and ~48% to 55% identical), including the presence of a unique cysteine residue in Mtb FtsZ (27). As such, it is possible that BPD compounds could specifically target mycobacterial FtsZ. However, given the still relatively high conservation of bacterial FtsZ, it is intriguing to speculate that BPD compounds may have a unique target in mycobacteria.

To determine the efficacy of BPD-6 and -9 in the direct killing of Mtb, we used an auto-luminescent Mtb mc²6206 strain (Mtb-*lux*) (28) to accelerate the measurement of bacterial viability upon treatment with BPD compounds. This recombinant Mtb-*lux* strain expresses a bacterial luciferase operon that allows the bacteria to generate a constitutive luminescence signal (29, 30). Luminescence signal not only depends on the proteins of the *lux* operon but also requires ATP and NADPH. As such, the relative luminescence unit (RLU) acts as a surrogate reporter for the viability of metabolically active Mtb and has been shown to correlate with CFU (29). Treatment of Mtb-*lux* with BPD-6 and -9 for 24 h

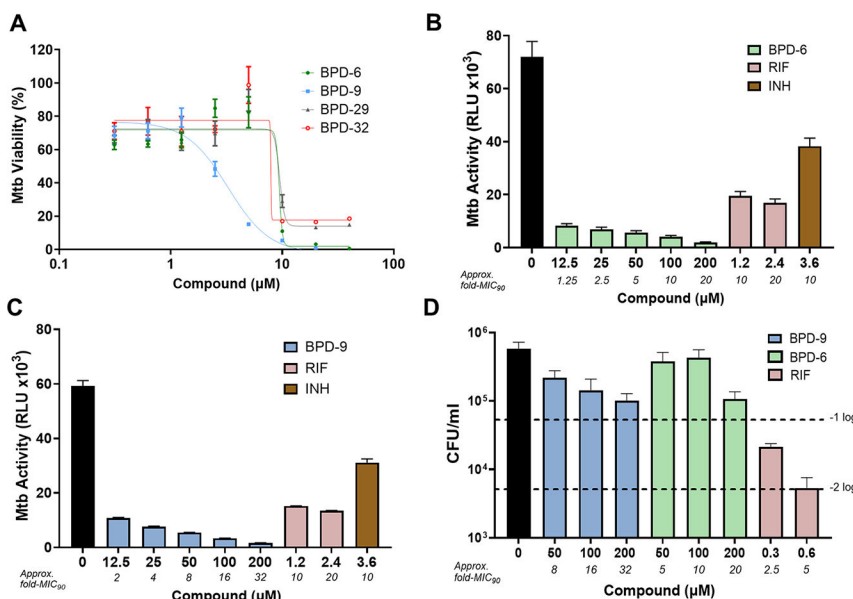

**FIG 2** Benzo[c]phenanthridine derivative (BPD)-6 and BPD-9 significantly inhibit *Mycobacterium tuberculosis* (Mtb) growth. (A) Dose-dependent activity of BPD-6, -9, -29, and -32 against Mtb mc$^2$6206 was determined using the resazurin microtiter assay (REMA). Mtb viability is normalized to maximal bacterial growth in the absence of compounds as 100%. (B, C) Mtb-*lux* was treated with BPD-6 (B) or BPD-9 (C) and rifampicin (RIF)/isoniazid (INH) as controls at the indicated concentrations for 20 h. The viability (activity) of Mtb-*lux* was determined by measuring the resulting luminescence of the strains (RLU, relative light unit). (D) Mtb mc$^2$6206 was mock treated or treated with BPD-6, BPD-9, and RIF for 20 h. Bacteria were serially diluted, and select dilutions were inoculated on 7H10 agar plates for enumeration of colony forming units (CFU). Data represent the mean ± SEM of three independent replicates.

resulted in a dose-dependent inhibition of Mtb viability (Fig. 2B and C). As a comparison for the activity of BPD compounds, we used the two key frontline TB antibiotics, RIF and isoniazid (INH), at relatively high concentrations (~10-fold MIC$_{90}$). Importantly, when used even at 1- to 2-fold their MIC$_{90}$, both BPD-6 and -9 showed increased inhibition of Mtb viability compared to RIF (1.2 µM) or INH (3.6 µM), which are 10-fold their respective MIC$_{90}$ (31). These results indicate that the two compounds function rapidly to potently inhibit the metabolism and replication of Mtb. However, the use of luciferase as a readout for viability has limitations, as bacteria with low metabolic activity may only produce non-detectable levels of luminescence signal. To assess whether the new BPD compounds possessed bactericidal ability, we performed the classical colony-forming unit (CFU) plating assay. Here, we used RIF as a known bactericidal TB antibiotic control at its minimum bactericidal concentration (MBC$_{90}$), which is approximately 5-fold its MIC$_{90}$ (32). Mtb treated with RIF showed the expected ~2-log (99%) reduction in bacteria CFU (Fig. 2D). However, both BPD-6 and -9 were unable to reach this level of efficacy even at 10-fold their MIC$_{90}$ (Fig. 2D). Although BPD-6 and -9 can kill Mtb as shown by a near 1-log reduction in CFU at high concentrations of >100 µM (Fig. 2D), it is clear that their bactericidal potency is limited. Instead, this suggests that BPD-6

**TABLE 1** MIC$_{90}$ of BPD-6, BPD-9, and SG against various bacterial species[a]

| Compound (µM) | *Mycobacterium* spp. | | | | Other bacterial species | | | | |
|---|---|---|---|---|---|---|---|---|---|
| | Mtb mc$^2$6206 | *M. bovis* BCG | *M. kansasii* | *M. smegmatis* | EC | LM | PA01 | PA14 | SE |
| BPD-6 | 10.12 ± 0.05 | 2.39 ± 0.13 | 7.58 ± 0.17 | 4.13 ± 1.63 | >40 | >40 | >40 | >40 | >40 |
| BPD-9 | 6.00 ± 1.02 | 2.26 ± 0.11 | 31.48 ± 3.42 | 2.63 ± 0.59 | >40 | >40 | >40 | >40 | >40 |
| SG | 74.21 ± 0.90 | 31.98 ± 1.96 | >40 | 31.61 ± 6.99 | ≤24[17] | n.d. | 99[19] | n.d. | n.d. |

[a]n.d., not determined; EC, *Escherichia coli*; LM, *Listeria monocytogenes*; PA, *Pseudomonas aeruginosa*; SE, *Salmonella enterica* Typhimurium.

and -9 act in a bacteriostatic manner. However, this does not discount the therapeutic potential of the compounds. For example, ethambutol (EMB), which acts through a bacteriostatic mechanism (33), is part of the regimen to treat active TB. As well, there is evidence that bacteriostatic macrolides have potential in the treatment of MDR-TB (34–36). Indeed, treatment of TB relies on multiple antibiotics to circumvent drug resistance, and bacteriostatic compounds are often combined with bactericidal drugs to achieve the best effect (37).

## BPD-6 and BPD-9 are active against non-replicating Mtb

Despite the lack of bactericidal activity, the BPD's ability to inhibit the metabolic activity of Mtb is comparable to or better than RIF. Bacteriostatic drugs targeting Mtb are valuable if they can also inhibit non-replicating bacteria, which have increased tolerance to most antibiotics (37). To determine the efficacy of BPD-6 and BPD-9 against Mtb with low metabolic activity, we used the established low-pH model to generate NR-Mtb (7). Mtb-*lux* was cultured in acidic phosphate citrate buffer without carbon or nitrogen supplements, and the resulting NR-Mtb-*lux* was used to assess the ability of the BPD compounds to inhibit their growth. The generated NR-Mtb-*lux* showed very low levels of luciferase activity (Fig. 3A), which supports that the bacteria are in a metabolically low, non-replicating state (38, 39). NR-Mtb-*lux* was incubated in complete media in the absence or presence of BPD compounds, or RIF and INH (at 10- to 20-fold their $MIC_{90}$) as controls for 20 h. Untreated NR-Mtb-*lux* recovered their ability to produce a luminescence signal, indicating their metabolic resuscitation and growth in this period of time (Fig. 3B and C). In contrast, BPD-6- and BPD-9-treated NR-Mtb-*lux* were completely inhibited from metabolic resuscitation and growth, to levels comparable or even better than RIF used at a relatively high concentration (10- to 20-fold $MIC_{90}$) (Fig. 3B and C). Although RIF has been shown to have good inhibitory activity against NR-Mtb (40), INH has only limited efficacy against NR-Mtb *in vitro* (7). Indeed, in the same set of experiments, treatment of NR-Mtb-*lux* with a high concentration of INH (10-fold $MIC_{90}$) did not show any significant inhibition of growth (Fig. 3B and C).

In physiological conditions, Mtb has the ability to maintain a dormant state within macrophages and reactivate upon perturbations to the immune system (41). Our *in vitro* NR-Mtb-*lux* model simulates this dormancy state, verified by the low level of luminescence (Fig. 3A). This luminescence, controlled by the *lux* operon, exploits fatty aldehydes as substrates to produce fatty acids and light, meaning, luminescent signal is dependent on an aerobic and metabolically active environment (29). In our experiments, we observed resuscitation of untreated NR-Mtb-*lux* upon incubation in complete growth media, due to restoration of metabolic activity and fatty acid biosynthesis (42). However, BPD-treated bacteria remained in a low metabolic state. Due to assay limitations in that growth or metabolic activity can only be measured in complete

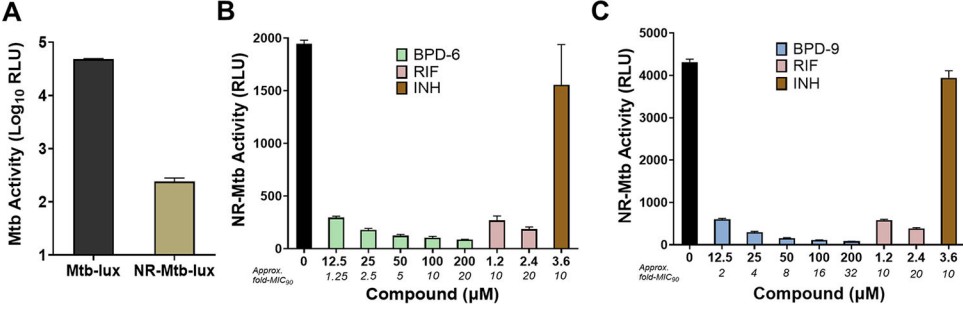

**FIG 3** Benzo[c]phenanthridine derivative (BPD)-6 and BPD-9 are active against non-replicating *Mycobacterium tuberculosis* (NR-Mtb). (A) Log-phase Mtb-*lux* and NR-Mtb-*lux* were resuspended to an $OD_{600}$ of 0.03 in 7H9 media, and their luminescence was measured immediately. (B, C) NR-Mtb-*lux* was treated with BPD-6 (B) or BPD-9 (C) and rifampicin (RIF)/isoniazid (INH) as controls at the indicated concentrations for 20 h. The activity of the compounds against NR-Mtb-*lux* was then determined by measuring the resulting luminescence (RLU). Data represent the mean ± SEM of three independent replicates.

media, we cannot conclude whether BPD compounds may be active in directly killing NR-Mtb-*lux*. However, considering that BPDs act in a bacteriostatic manner (Fig. 2D), it is likely that their target and mechanism of inhibiting NR-Mtb-*lux* are different from that of RIF. Many drug targets have been identified and well studied in Mtb with low metabolic activity, although none are specifically involved in the process of reactivation (6). Given that the BPD compounds prevent resuscitation of NR-Mtb-*lux*, the identification of a potentially novel drug target involved in the regulation of metabolic activity in NR-Mtb-*lux* is an appealing topic for future research. The question of whether BPD compounds can directly kill NR-Mtb, while possible and enticing, would require further experimentation.

## BPDs exhibit reduced cytotoxicity relative to SG and are effective against intracellular Mtb

The therapeutic potential of SG is limited given its cytotoxic properties due to non-specific targeting of essential eukaryotic proteins such as the $Na^+/K^+$-ATPase (43). Indeed, we show that human THP-1 macrophages treated with SG show significant cytotoxicity, with an $IC_{50}$ of ~9.5 µM (Fig. 4A). The $IC_{50}$ of SG is 5-fold lower than the $MIC_{50}$ of SG against Mtb, rendering it useless for targeting intracellular Mtb, which is critical given that Mtb is an intracellular pathogen. In contrast, BPD-6 and BPD-9 showed a cytotoxicity $IC_{50}$ of 43 and 15 µM in macrophages, respectively (Fig. 4A). As such, BPD-9 and BPD-6 have a cytotoxic $IC_{50}$ that is approximately 5-fold higher than their respective $MIC_{50}$ against Mtb. Taken together, the *in vitro* therapeutic ratio, defined as $IC_{50}/MIC_{50}$ for the respective compounds, illustrates the potential and improvement of BPD-6 and -9 compared to SG (Table 2). Despite their similar chemical structures, BPD-9 showed increased cytotoxicity compared to BPD-6, suggesting that a unique structural feature of BPD-9 contributes to its increased cytotoxicity. We speculate that minimizing the structural complexity of the BPD compounds will be key to reducing cytotoxicity while retaining activity, which is needed for translational potential of these compounds given that the therapeutic ratio is still relatively low.

The reduced cytotoxicity of the BPD compounds relative to SG enabled the evaluation of anti-Mtb efficacy in macrophage infection models. Mtb-*lux* has proven to be a reliable strain for determination of intracellular Mtb viability *in vitro* and *in vivo* (7, 29). Treatment of infected macrophages with BPD-6 and BPD-9 at 10 µM, which represents ~1 × $MIC_{90}$ and, importantly, is below their cytotoxic $IC_{50}$, effectively inhibited the intracellular growth of Mtb by 50% compared to untreated control cells (Fig. 4B and C). BPD-9 showed better inhibition than BPD-6, starting at 5 µM. These results indicate that the BPD compounds remain effective at inhibiting Mtb replication inside

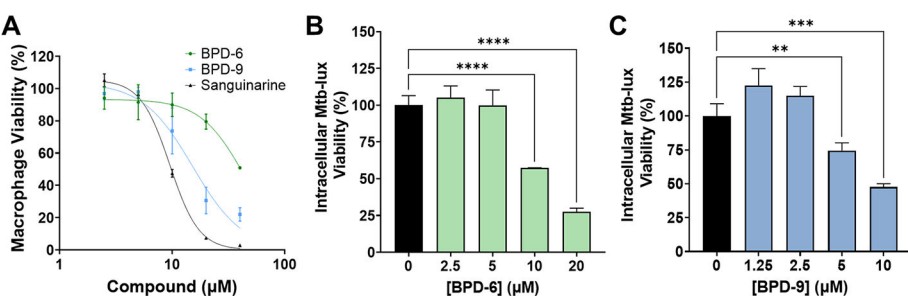

**FIG 4** Benzo[c]phenanthridine derivative (BPD)-6 and BPD-9 inhibit intracellular *Mycobacterium tuberculosis* (Mtb). (A) The cytotoxicity of BPD-6, BPD-9, and sanguinarine against THP-1 macrophages was determined at the indicated concentrations for 24 h. Macrophage viability was normalized relative to an untreated control as 100%. (B, C) THP-1 macrophages were infected with Mtb-*lux* at a multiplicity of infection (MOI) of 10. Infected cells were treated with BPD-6 (B) or BPD-9 (C) at the indicated concentrations for 24 h. The viability of intracellular Mtb was then determined by measuring the resulting luminescence (RLU). Data represent the mean ± SEM of three independent replicates. **$P < 0.01$; ***$P < 0.001$; ****$P < 0.0001$.

**TABLE 2** Therapeutic ratio (TR) of BPD-6, BPD-9, and SG

|  | Cytotoxicity $IC_{50}$ (µM) | Mtb $MIC_{50}$ (µM) | TR ($IC_{50}/MIC_{50}$) |
| --- | --- | --- | --- |
| BPD-6 | $42.56 \pm 3.45$ | $9.32 \pm 0.05$ | 4.57 |
| BPD-9 | $14.69 \pm 3.60$ | $2.94 \pm 0.39$ | 5.0 |
| SG | $9.46 \pm 0.55$ | $44.95 \pm 1.41$ | 0.21 |

the macrophage at near-MIC concentrations, which highlights the value of BPD-6 and -9 as novel compounds that have potential to treat TB.

### BPD-9 is active *in vivo* against *M. bovis* BCG

To further evaluate the toxicity and explore the therapeutic potential of BPD compounds, we performed *in vivo* analyses in mice. Based on a more favorable *in vitro* efficacy and therapeutic ratio (Table 2), we decided to test BPD-9. Daily intraperitoneal (i.p.) administrations of BPD-9 are well tolerated by mice up to a dose of 5 mg/kg for at least 14 d, as shown by the body weight score (Fig. 5A). The highest tested dose (10 mg/kg) showed a body weight loss of ~15% after the first few administrations, limiting the safe dosage to 5 mg/kg when delivered i.p. (Fig. S4). We then tested the efficacy of BPD-9 *in vivo* in a model of *M. bovis* BCG infection. We chose to use *M. bovis* BCG as a model because it has relatively greater replication competency and susceptibility to BPD-9 compared to the auxotroph $mc^2 6206$ Mtb strain (44, 45) (Table 1). Mice were infected by intravenous (i.v.) injection with ~$10^7$ CFU *M. bovis* BCG, and BPD-9 was administered at 5 mg/kg via i.p. daily for 8 d post-infection before performing CFU analysis of the organs. As shown in Fig. 5B, although no differences in CFU were observed in the spleen, the BPD-treated mice had significantly lower bacteria burden in the lungs, indicating that this compound retains its anti-mycobacterial activity *in vivo*. Despite that further work will be required to characterize the *in vivo* efficacy of BPD-9 against virulent Mtb, the results, so far, are promising.

### Combination treatment of BPD compounds with rifampicin improves anti-Mtb activity in axenic conditions and in infected macrophages

The combinatorial effectiveness of different antibiotics and drugs is one of the most emphasized aspects in drug development. To evaluate possible combinatorial effects of BPD-6 and BPD-9 with existing TB antibiotics, we adopted the combination checkerboard assay (46). This assay can be used to evaluate drug synergy through the calculation of the fractional inhibitory concentration index (FIC) (47). Checkerboard experiments did not show synergistic effects between either BPD compound and RIF, INH, EMB, or moxifloxacin (MFX) as FICs were >0.5 (data not shown). Importantly, there

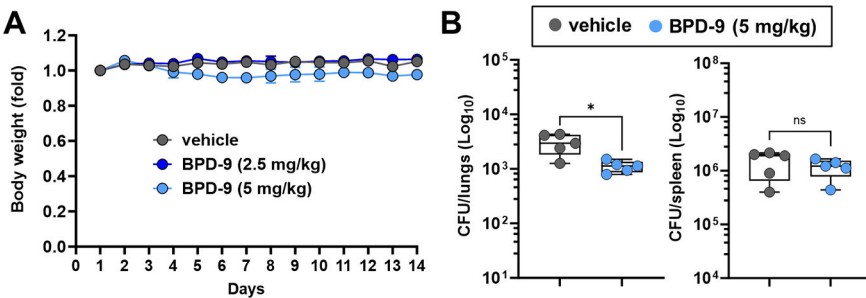

**FIG 5** Benzo[c]phenanthridine derivative (BPD)-9 treatment inhibits the *in vivo* growth of *M. bovis* BCG. (A) BALB/c mice (*n* = 3) were intraperitoneally (i.p.) injected with BPD-9 daily, up to 14 d. The animals were monitored for any sign of toxicity, and the mouse body weight was recorded before each injection. (B) Lungs and spleen CFU analysis of BALB/c mice (*n* = 5) infected with *M. bovis* BCG (~$10^7$ CFU, intra-vein) and treated daily via i.p. injections with BPD-9 at 5 mg/kg, for 8 d post-infection. Each dot represents one mouse. The statistical analysis was performed using Student's *t*-test. ns, not significant. *$P < 0.05$.

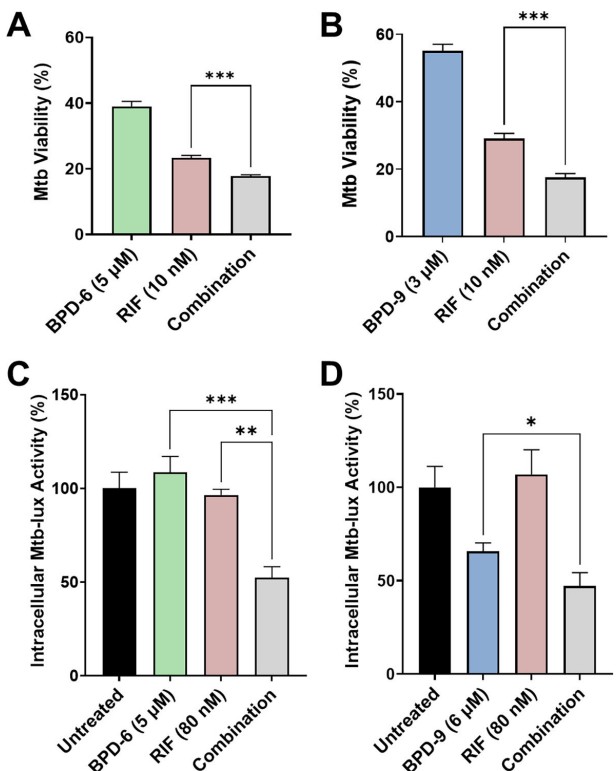

**FIG 6** Benzo[c]phenanthridine derivative (BPD)-6 and BPD-9 combine with rifampicin (RIF) to improve anti-*Mycobacterium tuberculosis* (Mtb) activity. (A, B) Activity of BPD-6 (A) or BPD-9 (B) alone or in combination with RIF against Mtb mc$^2$6206 was determined using the resazurin microtiter assay (REMA). Mtb viability is normalized to maximal bacterial growth in the absence of compounds as 100%. (C, D) THP-1 macrophages were infected with Mtb-*lux* and treated with BPD-6 (C) or BPD-9 (D) alone or in combination with RIF for 24 h. Intracellular Mtb viability was determined by measuring the resulting luminescence. Data represent the mean ± SEM of three independent replicates. *$P < 0.05$; ***$P < 0.001$.

were no antagonistic effects with any of the tested TB drugs. However, these assays revealed potential additive effects between BPD-6 and -9 with RIF. To confirm these observations, we performed the REMA with Mtb treated with either BPD-6, -9, or RIF alone, or in combination at sub-MIC concentrations. These assays showed that the combined treatment of Mtb with BPD-6 or BPD-9 with RIF significantly decreased Mtb growth compared to single treatments (Fig. 6A and B). The combination effect was specific to RIF and did not occur in combination with INH, EMB, and MFX. Importantly, the increased activity of BPD-6 and -9 in combination with RIF translated to intracellular conditions, where we observed a decrease in Mtb viability in macrophages treated with the combination of BPD-6 or -9 and RIF compared to single treatments (Fig. 6C and D). Given that the combination effect with BPDs only occurred with RIF, we speculate that the mechanism of action for BPDs is distinct from RIF. Combination effects are desirable because it can lower the effective concentrations of each drug to minimize cytotoxicity.

## BPD-6 and BPD-9 accumulate within the bacterial cell

To gain insight into how the BPD compounds may exert their antibacterial functions, we sought to investigate the dynamics of the compounds' interaction with bacteria. Interestingly, BPD-6 and BPD-9 have fluorescence properties (peak excitation/emission wavelengths of 420/485 nm) that can be exploited for visualization and quantification (Fig. S5). To examine the uptake and localization of the compounds inside the bacteria, Mtb-RFP (48) treated with sub-MIC concentrations of BPD-6 and BPD-9 were visualized by epifluorescence microscopy. Fluorescence of BPD-6 and BPD-9 co-localized with

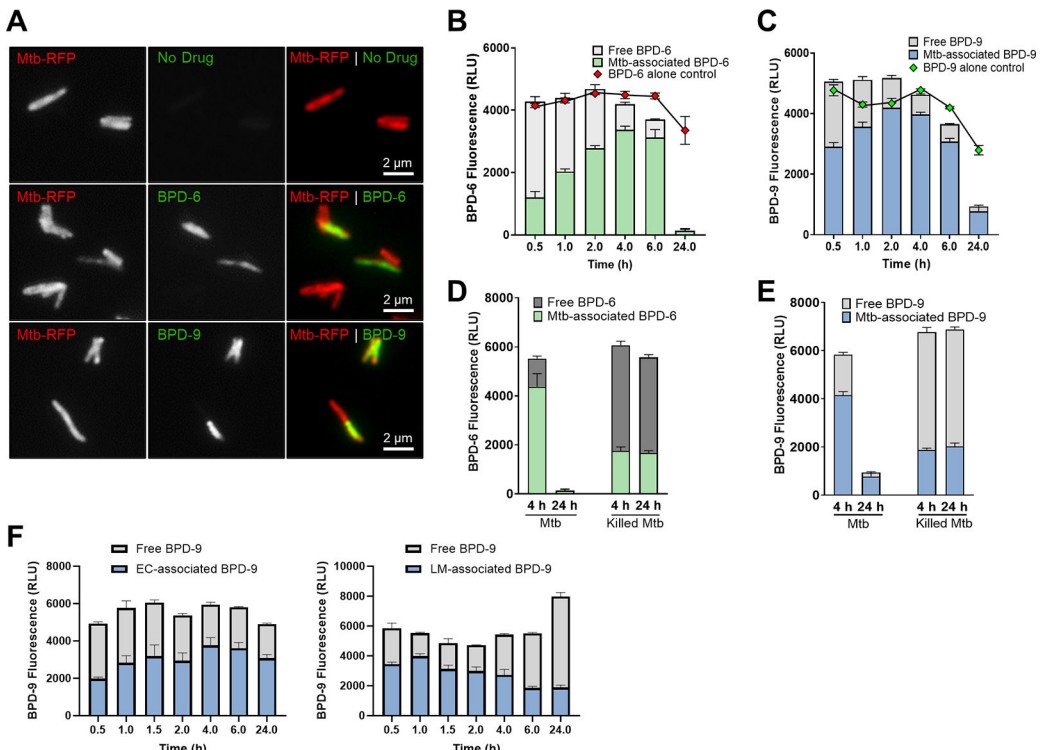

**FIG 7** Benzo[c]phenanthridine derivative (BPD)-6 and BPD-9 accumulate within *Mycobacterium tuberculosis* (Mtb) cells. (A) Representative fluorescence image of Mtb-RFP treated with 5 µM BPD-6 or BPD-9 for 4 h. (B, C) Mtb mc²6206 was treated with 5 µM BPD-6 (B) or BPD-9 (C) for the indicated time points. The resulting fluorescence of the BPD compounds in the Mtb pellet (Mtb-associated BPD) and the supernatant (free BPD) was measured. A "BPD-alone control" in the absence of Mtb was included to measure the natural decay of BPD's fluorescence. (D, E) Live or gentamicin-killed Mtb mc²6206 were treated with 5 µM BPD-6 (D) or BPD-9 (E) for 4 and 24 h, and the accumulation of the compounds in Mtb was measured by the resulting fluorescence. (F) *E. coli* (EC) and *L. monocytogenes* (LM) were treated with 5 µM BPD-9 for 24 h, and the accumulation of the compounds was measured by fluorescence. Data represent the mean ± SEM of three independent replicates.

RFP signal and was evenly distributed throughout the bacteria, indicating a possible accumulation of the compounds in the cytosol of Mtb cells (Fig. 7A). However, some of the Mtb cells in the sample did not present a fluorescence signal, which could be explained either by heterogeneity in the uptake of the compound or possibly in the degradation and/or efflux of the compound.

To quantify the accumulation of the compounds bound to or inside the bacteria over a period of 24 h, we measured fluorescence in the Mtb pellet (Mtb associated) and the supernatant (free), which would contain a compound that was either not taken up or effluxed by the bacteria. Maximal uptake of BPD-6 and BPD-9 occurred within 4 h of treatment, which corresponded to a relatively low percent of unassociated compound remaining in the supernatant (Fig. 7B and C). This ability of the BPD compounds to accumulate within the Mtb is consistent with the chemical properties of benzo[c]phenanthridine compounds, which tend to be nonpolar with a high level of molecular planarity, enabling them to more easily enter the bacterial cell (49). Interestingly, we observed a dramatic decrease in fluorescence in the Mtb pellet at 24 h post-treatment (Fig. 7B and C). The magnitude of this decrease in fluorescence is significantly higher than the natural decrease observed when the compounds are incubated in the exact same conditions without Mtb. There was also no evidence of increased efflux in the supernatant. A possible explanation for the decrease in fluorescence of the compounds following prolonged incubation with Mtb is that the bacteria are able to metabolize or modify the compound, or that interaction with its cognate target within Mtb interferes with the intrinsic fluorescence of the compound. Consistent with the hypothesis of

compound metabolizing, we show that only treatment of live, but not killed Mtb (Fig. S6), results in a decrease in fluorescence at 24 h post-treatment (Fig. 7D and E). In addition, metabolically active live bacteria are required for uptake of the compound as killed bacteria accumulate significantly lower amounts of both BPD-6 and BPD-9 after 4 h of treatment (Fig. 7D and E). This may help to explain the heterogeneous presentation of the compound accumulation observed in Fig. 7A as different cells within the population could be at different stages of growth with different metabolic states. It is possible that the cells lacking BPD fluorescence signal were at a stage of growth with lower metabolic activity and, therefore, had slower uptake of the compound. Conversely, it is also possible that these Mtb were at a stage of growth with relatively increased metabolic activity and had already begun to metabolize or modify the compound leading to loss of signal.

To further explore the mechanism behind the selectivity of BPD-9 to mycobacteria, we also investigated the association of the compound to both *E. coli* and *Listeria monocytogenes*, which are not affected by BPD-9 (Table 1). As with Mtb, we measured the fluorescence in both the supernatant and bacterial pellet following incubation. In both *E. coli* and *L. monocytogenes*, BPD-9 accumulated in the pellet following 60–90 minutes of incubation (Fig. 7F). In the *E. coli* pellet, the signal remained high even after 24 h and there was no sharp decrease like what was observed for Mtb (Fig. 7C). The *L. monocytogenes* pellet showed not only a decrease in fluorescence after 6 h but also a corresponding increase in the supernatant fluorescence (Fig. 7F), which was not observed for Mtb (Fig. 7C). These results suggest that BPD-9 does associate to both *E. coli* and *L. monocytogenes*, but is not metabolized by those bacteria and may even be effluxed more efficiently, possibly explaining, at least in part, why BPD-9 is inactive against these species.

## BPD-6 and BPD-9 are effective against virulent and clinical MDR Mtb strains

Given the increasing prevalence of MDR-TB, novel anti-Mtb compounds that retain their efficacy against clinical and MDR strains of Mtb are highly desirable. We show that both BPD-6 and BPD-9 retain their efficacy in the low micromolar MIC range against both virulent laboratory strains (Mtb H37Rv and Mtb Erdman) and the hypervirulent Mtb HN878 strain (Table 3; Fig. S7A-C). Importantly, we examine the efficacy of both compounds against a panel of five clinical drug-resistant Mtb isolates (50: INH^R; 105: PZA^R, INH^R, STR^R; 116: INH^R, RIF^R, STR^R; 151: PZA^R, INH^R, RIF^R, STR^R; 217: PZA^R, INH^R, RIF^R, ETO^R). Strikingly, both BPD-6 and BPD-9 effectively inhibited the growth of all five clinical MDR Mtb strains in the low micromolar range, whereas resistance to rifampicin and isoniazid was observed in the corresponding isolates (Table 3; Fig. S7D-H). Collectively, our data show that BPDs are effective against multiple strains of virulent and MDR Mtb, and that their mechanism of action is distinct from that of the clinically relevant TB drugs, including rifampicin, isoniazid, pyrazinamide, streptomycin, and ethionamide.

## Mtb Mmr efflux pump contributes to resistance against BPD-9

To determine the mechanism of action of BPD-9 against Mtb, we performed a drug-stressing genetic screen with BPD-9 on Mtb H37Rv. We isolated four BPD-9-resistant strains and performed whole genome sequencing to identify gene(s) with mutations. All four strains (R1_BPD-9, R2_BPD-9, R3_BPD-9, and R4_BPD-9) showed a mutation in *rv3066* (Table 4), which encodes a transcriptional repressor that can negatively regulate the expression of the small multidrug efflux pump Mmr, encoded by *rv3065* (50, 51). In addition to the *rv3066* mutation, we observed additional mutations in all four BPD-9-resistant mutants. In R2_ and R4_BPD-9, we observed an Arg51Gly mutation in *rv1960*, which encodes for a possible antitoxin ParD1, part of the type II toxin-antitoxin system (52). In R1_ and R3_BPD-9 mutants, we observe two non-synonymous mutations in *rv2962c* (Thr234Ile) and *rv2933* (Glu898fs). These genes encode for a possible glycosyl transferase implicated in intermediary metabolism and respiration (53), and a phenolphthiocerol synthesis type-I polyketide synthase PpsC involved in the biosynthesis of phenolphthiocerol and phthiocerol dimycocerosate (PDIM) (54), respectively.

**TABLE 3** MIC of sanguinarine (SG), BPD-6, BPD-9, rifampicin (RIF), and isoniazid (INH) against various laboratory strains and clinical isolates of *M. tuberculosis*[a]

| Strains | Drug resistance | MIC (µM) | | | | | | | | | |
| --- | --- | --- | --- | --- | --- | --- | --- | --- | --- | --- | --- |
| | | SG | | BPD-6 | | BPD-9 | | RIF | | INH | |
| | | MIC50 | MIC90 | MIC50 | MIC90 | MIC50 | MIC90 | MIC50 | MIC90 | MIC50 | MIC90 |
| H37Rv | None | >10 | >10 | 11.4 ± 0.3 | 12.6 ± 0.1 | 6.7 ± 0.9 | 8.2 ± 2.5 | 0.003 ± 0.00002 | 0.003 ± 0.00002 | 0.4 ± 0.001 | 0.4 ± 0.007 |
| Erdman | None | >10 | >10 | 9.3 ± 3.14 | 14.3 ± 5.5 | 7.7 ± 1.8 | 11.7 ± 5.3 | 0.006 ± 0.002 | 0.01 ± 0.004 | 0.7 ± 0.1 | 0.7 ± 0.2 |
| HN878 | None | >10 | >10 | 8.8 ± 2.9 | 9.5 ± 3.1 | 4.6 ± 1.2 | 5.9 ± 1.2 | 0.003 ± 0.0005 | 0.004 ± 0.001 | 0.4 ± 0.05 | 0.6 ± 0.2 |
| 50 | INH | >10 | >10 | 8.3 ± 2.5 | 10.0 ± 2.7 | 5.2 ± 1.1 | 6.7 ± 0.1 | 0.003 ± 0.001 | 0.005 ± 0.002 | 3.0 ± 0.1 | 5.6 ± 2.3 |
| 105 | PZA, INH, other, STR | >10 | >10 | 5.1 ± 0.9 | 9.2 ± 2.8 | 3.5 ± 0.6 | 5.2 ± 1.8 | 0.001 ± 0.0003 | 0.003 ± 0.0004 | ≥100 | ≥100 |
| 116 | INH, RIF, STR | >10 | >10 | 4.4 ± 0.8 | 7.7 ± 0.8 | 3.4 ± 0.8 | 5.6 ± 2.0 | 48.2 ± 5.1 | 58.4 ± 4.8 | 7.9 ± 0.2 | 12.6 ± 0.6 |
| 151 | PZA, INH, other, RIF, STR | >10 | >10 | 2.4 ± 0.1 | 3.5 ± 0.1 | 1.8 ± 0.2 | 2.9 ± 0.4 | 24.8 ± 10.5 | 52.3 ± 12.7 | 27.1 ± 2.2 | 36.2 ± 9.6 |
| 217 | PZA, INH, RIF, ETO | >10 | >10 | 4.3 ± 1.4 | 4.8 ± 1.4 | 2.9 ± 0.06 | 3.2 ± 0.06 | 15.2 ± 5.9 | 17.1 ± 4.1 | ≥100 | ≥100 |

[a]PZA, pyrazinamide; STR, streptomycin; ETO, ethionamide.

**TABLE 4** Drug-stressing genetic screen with BPD-9 on Mtb H37Rv[a]

| Genes | Gene annotation | Resistant mutant (MIC$_{90}$) | | | |
|---|---|---|---|---|---|
| | | R1_BPD-9 (50.6 ± 0.5 µM) | R2_BPD-9 (55.2 ± 3.6 µM) | R3_BPD-9 (51.8 ± 1.1 µM) | R4_BPD-9 (50.2 ± 0.05 µM) |
| *rv1960* | Possible antitoxin ParD1 | | C152G (p.Arg51Gly) | | C152G (p.Arg51Gly) |
| *rv2678c* | Probable uroporphyrinogen decarboxylase HemE | C728T (p.His243His) | | | |
| *rv2933* | Phenolphthiocerol synthesis type-I polyketide synthase PpsC | A2685AC (p.Glu898fs) | | A2685AC (p.Glu898fs) | |
| *rv2962c* | Possible glycosyl transferase | C700T (p.Thr234Ile) | | C700T (p.Thr234Ile) | |
| *rv3066* | Probable transcriptional regulatory protein (probably DeoR family) | G44T (p.Arg15Leu) | G469A (p.Gly157Ser) | C345G (p.Tyr115*) | T197C (p.Leu66Pro) |

[a]Four resistant mutants were isolated for whole genome screening, with mutations in the indicated genes. MIC$_{90}$ (µM) values are shown below each strain in parenthesis.

Interestingly, in all four BPD-9-resistant mutants, none of the *rv3066* mutations are within its DNA binding motif (aa36-55). However, the four mutant clones all show >100-fold increased expression of *mmr* compared to H37Rv (Fig. 8A) with MIC$_{90}$ values ranging from 50.2 to 55.2 µM (Table 4). The marked increase in *mmr* expression in the BPD-9-resistant mutants is also comparable to levels found in isogenic deletion mutants of *rv3066* in the H37Rv strain (Δ*rv3066* strains; Fig. 8A). Notably, Δ*rv3066* strains also show increased resistance to BPD-9, likely due to the overexpression of Mmr, as deletion of the entire operon (Δ*mmr-rv3066*) restores susceptibility to BPD-9 (Fig. 8B and D). Deletion of either *rv3066* or the *mmr-rv3066* operon does not affect rifampicin resistance (Fig. 8C and D), suggesting that Mmr-mediated resistance is specific to BPD-9 and does not impact sensitivity to rifampicin. Taken together, these results suggest that transcriptionally regulated overexpression of *mmr* provides the first line of resistance for Mtb to BPD-9 and that the other mutations observed in the four BPD-9-resistant mutants are not contributing to BPD-9 resistance. As such, the existence of the Mmr efflux system occludes the discovery of the direct target of BPD-9 in mycobacteria, and further work is needed to identify the specific target of BPD-9.

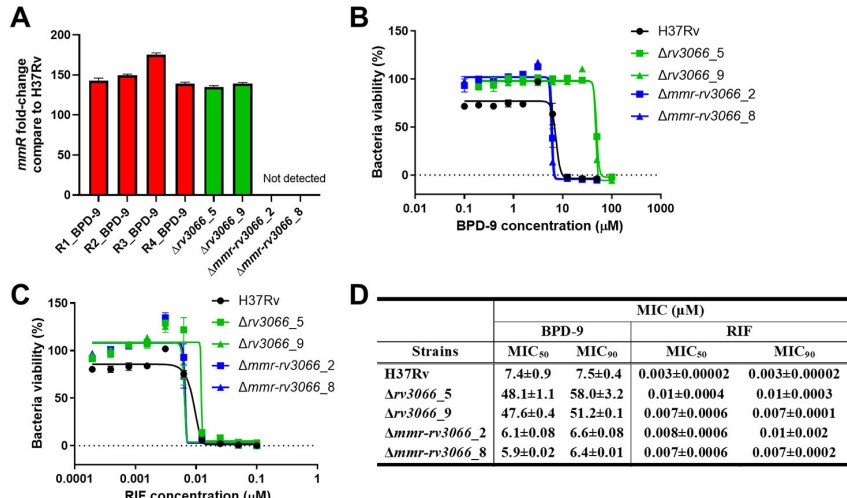

**FIG 8** The Mmr efflux pump is involved in resistance to benzo[c]phenanthridine derivative (BPD)-9. (A) The expression of the *mmr* gene of four BPD-9-resistant mutants and two Δ*rv3066* mutants was compared to that of H37Rv. (B, C) Two Δ*rv3066* mutant clones and two Δ*mmr-Rv3066* mutant clones along with H37Rv were treated with BPD-9 (B) or rifampicin (C), and bacterial viability was assessed with the resazurin microtiter assay (REMA). The corresponding MIC values are shown in (D). Data are represented as the mean ± standard deviation (SD) of four replicates.

## Conclusions

In summary, we report the design and synthesis of a group of benzo[c]phenanthridine compounds derived from sanguinarine that possesses potent and specific anti-mycobacterial activity. Our data also provide evidence for the active pharmacophore of benzo[c]phenanthridine compounds and the moieties that impact cytotoxicity and activity. We provide biological evidence that these compounds inhibit metabolically inactive non-replicating Mtb, intracellular Mtb, and multiple clinical MDR-TB strains in the low micromolar concentration range. The potency, specificity, and cytotoxicity are all superior relative to sanguinarine, and suggest a unique mode of action that is distinct from all frontline TB antibiotics and which likely involves the Mmr efflux pump. The selective ability to inhibit multiple species of *Mycobacterium* also supports a target that is unique to this genus. BPD-9 retained antibacterial activity against *M. bovis* BCG in mice, showing that BPD compounds have the ability to act *in vivo* and, therefore, have good therapeutic potential. Although more work is needed to identify the specific target, the information gained thus far will be valuable for the development of more potent antibiotics that specifically target mycobacteria.

## MATERIALS AND METHODS

### Chemistry

#### General procedures

Starting materials, reagents, and solvents were purchased from commercial suppliers and used without further purification. Silica gel for column chromatography was of 200–300 mesh particle size, EA/PE mixture and DCM/MeOH mixture were used for purification. NMR spectra were recorded at room temperature on Bruker Avance III 400 Spectrometer (400 MHz). Chemical shifts are given in parts per million and coupling constants in hertz. $^1$H spectra were calibrated in relation to the reference measurement of TMS (0.00 ppm). $^{13}$C spectra were calibrated in relation to deuterated solvents. High-resolution mass spectra (HRMS) were performed using the Agilent G6520 Q-TOF high-resolution mass spectrometer. The purity of active compounds BPD-6, -9, -19, -29, and -32 was determined by high-performance liquid chromatography (HPLC). HPLC conditions: Agilent 1260 with an XBridge C18 column (4.6 mm × 150 mm, 5 µm); T = 40°C; flow rate = 0.5 mL/min. All final compounds were confirmed to have a purity of over 95%.

#### Synthetic route for BPD-6 and BPD-9

For the synthesis of the most potent compounds BPD-6 and BPD-9 (Fig. 9), an F-C acetylation was utilized starting from anisole and veratrole, respectively. Treatment anisole or veratrole with succinic anhydride under $AlCl_3$ condition provided **A2** after workup. Next, **A2** was converted into **A4** by reduction with $Et_3SiH$ and dehydration in PPA, and aromatization was conducted by subsequently treating **A4** with $Br_2$ and DBU. The corresponding arylamine **A9** was prepared by $S_N2$, Smiles rearrangement, and hydrolysis reactions. Finally, **A9** was treated with formyl acetate, MeI, followed by a Suzuki coupling with aminophenyl borate and Vilsmeier-Haack reaction in $POCl_3$ condition for ring closure to yield BPD-6 and BPD-9. The detailed synthetic procedure are as follows:

#### Step a

To a mixture of **A1** (1.0 equiv) and succinic anhydride (1.25 equiv) in nitrobenzene (2.67 M), $AlCl_3$ (2.0 equiv) was added in ice bath, and the reaction mixture was stirred at room temperature and monitored by TLC. After completion of the reaction, the reaction was quenched by ice water and washed with water and EA. The organic phase was dried over $Na_2SO_4$ and concentrated under reduced pressure. The residue was recrystallized to afford **A2**.

**FIG 9** Synthetic route for benzo[c]phenanthridine derivative (BPD)-6 and BPD-9. Reagents and conditions: (a) $AlCl_3$, succinic anhydride, nitrobenzene, 60°C; (b) $Et_3SiH$, TFA, 100°C; (c) PPA, 80°C; (d) i. HBr, 125°C; ii. $CH_2Br_2$, $K_2CO_3$, DMF, 85°C; (e) $Br_2$, $CHCl_3$, r.t.; (f) DBU, MeCN, 50°C; (g) 2-bromo-2-methylpropanamide, NaOH, DMF, r.t.; (h) NaH, DMF/DMTP (4:1), 100°C; (i) NaOH, MeOH/$H_2O$ (1:1), 100°C; (j) formyl acetate, THF, r.t.; (k) MeI, NaH, DMF, r.t.; (l) N,N-dimethyl-3-(4,4,5,5-tetramethyl-1,3,2-dioxaborolan-2-yl)aniline, Pd(PPh$_3$)$_4$, $K_3PO_4$, 1,4-dioxane/$H_2O$ (1:1), 90°C, $N_2$; (m) POCl$_3$, 50°C.

## Steps b and c

A mixture of **A2** (1.0 equiv) and $Et_3SiH$ (2.5 equiv) in TFA (1.4 M) was stirred at 100°C for 7 h. The reaction mixture was cooled to room temperature and then diluted with water. The product was extracted with EA, then the organic extract was dried over $Na_2SO_4$ and concentrated under reduced pressure. Polyphosphoric acid (0.5 mL/mmol) was added to the crude product dropwise at room temperature, and the reaction mass was heated at 80°C and monitored by TLC. After completion of the reaction, the mixture was diluted with EA in 0°C and quenched by ice water. The product was extracted with EA, and the combined organic phase was dried over $Na_2SO_4$ and concentrated under reduced pressure. The residue was purified by silica gel chromatography to afford **A4**.

## Step d

A mixture of **A4-2** in 48% HBr (0.325 M) was refluxed at 125°C for 6 h. The reaction was cooled to room temperature, and the brown crystal was filtered and dried to afford the crude product. The brown crystal was then diluted in DMF (0.2 M), and $K_2CO_3$ (5.0 equiv) and $CH_2Br_2$ (1.2 equiv) were added. The mixture was heated at 85°C for 2 h. After the reaction was cooled to room temperature, the solvent was removed and the residue was diluted with EA. The solution was washed with water, dried over $Na_2SO_4$, and purified by silica gel chromatography to afford **A4-3**.

## Steps e and f

To a mixture of **A4** in chloroform (1.5 M), $Br_2$ was added dropwise at room temperature, and the reaction was stirred at room temperature and monitored by TLC. After completion of the reaction, the mixture was quenched by aqueous $Na_2SO_3$, washed with water, and extracted with EA. The organic phase was dried over $Na_2SO_4$ and concentrated under reduced pressure. The crude product was dissolved in MeCN (1.0 M), DBU (1.5 equiv) was added at 50°C, and the mixture was then stirred at 50°C for 10 min. The

reaction was diluted with EA, washed with water, dried over $Na_2SO_4$ and concentrated under reduced pressure, then purified by silica gel chromatography to give **A6**.

### Step g

To a solution of **A6** (1.0 equiv) in DMF (0.2 M), NaOH (2.0 equiv) was added. After stirring at room temperature for 30 min, 2-bromo-2-methylpropanamide (2.0 equiv) was added and then stirred for 2 h at ambient temperature. The mixture was diluted with EA, washed with water and dried over $Na_2SO_4$, then concentrated under reduced pressure and purified by silica gel chromatography to give **A7**.

### Steps h and i

To a solution of **A7** in DMF/DMTP (4:1, 0.22 M), NaH (2.0 equiv) was added at room temperature, and the mixture was heated at 100°C for 4 h. The reaction was cooled to room temperature and quenched by aqueous $NH_4Cl$. The product was extracted with EA, dried over $Na_2SO_4$, concentrated under reduced pressure, and purified by silica gel chromatography to give **A8**. **A8** was then dissolved in $MeOH/H_2O$ (1:1, 0.15 M), and NaOH (wt/wt 80%) was added. The resulting mixture was then heated at 100 °C and monitored by TLC. After completion of the reaction, the mixture was washed with water, extracted with EA, concentrated under reduced pressure, and purified by gel chromatography to afford **A9**.

### Steps j and k

To a solution of **A9** (1.0 equiv) in THF (0.32 M), formyl acetate (3.0 equiv) was added and the mixture was stirred for 30 min at room temperature. The product was washed with water and extracted with EA. The organic phase was dried over $Na_2SO_4$ and concentrated under reduced pressure to afford **A10**. A mixture of **A10** (1.0 equiv) and NaH (2.0 equiv) in DMF (0.27 M) was stirred for 10 min at room temperature, then MeI (3.0 equiv) was added, and the reaction was stirred for 1 h at ambient temperature. After completion of the reaction, the mixture was washed with water and extracted with EA, dried over $Na_2SO_4$, and concentrated under reduced pressure to afford **A11**.

### Steps l and m

To a solution of **A11** (1.0 equiv) in 1,4-dioxane/$H_2O$ (1:1, 0.075 M), N,N-dimethyl-3-(4,4,5,5-tetramethyl-1,3,2-dioxaborolan-2-yl)aniline (1.5 equiv), $Pd(PPh_3)_4$ (5 mol%), and $K_3PO_4$ (1.5 equiv) were added under nitrogen atmosphere, and the suspension was heated at 90°C for 3 h. The reaction was cooled to room temperature, washed with water and extracted with EA, then concentrated under reduced pressure to give a yellow oil. Next, $POCl_3$ was added (1.0 mL/mmol) to the crude product, and the mixture was heated at 60°C and monitored by TLC. After completion of the reaction, the mixture was diluted with DCM, then a yellow solid appeared and was filtered to afford the product **BPD-6** and **BPD-9**. Total yield for BPD-6 and BPD-9 was 2.6% and 1.2%, respectively.

## Biology

### Cell culture

THP-1 monocytes (ATCC TIB-202) were maintained at 37°C in a humidified atmosphere of 5% $CO_2$ in RPMI 1640 media (Gibco, Gaithersburg, MD, USA) supplemented with 10% heat-inactivated fetal bovine serum (FBS), 10 mM HEPES, penicillin (100 IU/mL), streptomycin (100 µg/mL), and 2 mM L-glutamine purchased from Gibco. For differentiation into THP-1 macrophages, cells were resuspended in complete RPMI 1640 media (without antibiotics) and incubated with 100 ng/mL phorbol 12-myristate 13-acetate

(PMA, Alfa Aesar, Haverhill, MA, USA). The cells were seeded at 50,000 cells/well into 96-well plates and were incubated at 37°C for 3 d.

### Bacterial culture

A complete list of strains used in this study can be found in Table S1. The *Mycobacterium tuberculosis* H37Rv-derived auxotrophic strain mc$^2$6206 (Δ*panCD*Δ*leuCD*) (45), virulent strains (Mtb Erdman, H37Rv, HN878), clinical MDR strains (isolates #50, 105, 116, 151, 217), *M. kansasii* Hauduroy (ATCC 12478), and *M. bovis* BCG (strain Institute Pasteur, ATCC 27291) were cultured in Middlebrook 7H9 media (BD Biosciences, Franklin Lakes, NJ, USA) supplemented with 0.05% Tween 80 (Acros Organics, Fair Lawn, NJ, USA), 0.2% glycerol (Fisher Chemical, Waltham, MA, USA), and 10% OADC (BD Biosciences). For the Mtb mc$^2$6206 strain, D-pantothenic acid (24 µg/mL, Alfa Aesar) and L-leucine (50 µg/mL, Alfa Aesar) were also added. For *M. kansasii*, 5 µg/mL streptomycin was also added (Fisher Scientific, Waltham, MA, USA). *M. smegmatis* mc$^2$155 (ATCC 700084) was maintained in 7H9 media supplemented with 10% ADS enrichment: 50 g bovine serum albumin (VWR, Radnor, PA, USA), 20 g dextrose (Fisher Scientific), 0.85% (wt/vol) sodium chloride (Fisher Chemical), 0.2% glycerol, and 0.05% tyloxapol (Acros Organics). Clinical MDR Mtb isolates were obtained from the McGill International TB Centre (Montreal, Canada).

The recombinant bioluminescent strain Mtb-*lux* was generated as previously described (28) by transforming the pMV306hsp+LuxG13 plasmid into Mtb mc$^2$6206 (29, 30). pMV306hsp+LuxG13 was a gift from Brian Robertson and Siouxsie Wiles (Addgene plasmid # 26161; http://n2t.net/addgene:26161; RRID:Addgene_26161). This plasmid encodes a bacterial luciferase-expressing operon *lux* and G13 promoter, allowing the constant production of luminescence signal by metabolically active bacteria. The strain was maintained in complete 7H9 media supplemented with 30 µg/mL kanamycin (Fisher Scientific). Mtb mc$^2$6206 expressing tdTomato (Mtb-RFP) was generated previously (48) and maintained in complete 7H9 media supplemented with 50 µg/mL hygromycin.

*E. coli* strain NEB Stable (New England Biolabs, Ipswich, MA, USA), *P. aeruginosa* strains PA01 and PA14 (generous gift from Dr. Thien-Fah Mah, University of Ottawa), *Salmonella enterica* serovar Typhimurium strain SL1344, and *L. monocytogenes* strain 10403s (both generous gifts from Dr. Subash Sad, University of Ottawa) were cultured in lysogeny broth (LB, Fisher Scientific). Bacteria were inoculated into LB from frozen glycerol stocks and grown at 37°C with shaking (200 rpm).

### Resazurin microtiter assay (REMA)

Bacteria were diluted to an initial optical density (OD$_{600nm}$) of 0.02 at mid-log stage and were incubated with sanguinarine (Tocris Bioscience, Toronto, Canada), RIF (Fisher Scientific), INH (Acros Organics), EMB (Alfa Aesar), MFX (Alfa Aesar), or BPD compounds at 37°C. The incubation time was 5 d for Mtb and *M. bovis* BCG, 20 h for *M. smegmatis*, and 3 d for *M. kansasii*. Then, resazurin (Sigma-Aldrich, St. Louis, MO, USA) solution was added to the wells to achieve a final concentration of 100 µM resazurin and 0.5% Tween 80. The mixtures were incubated at 37°C before measurement of fluorescence with the Synergy H1 Microplate Reader (BioTek, Winooski, VT, USA). The readout was performed after 6 h for Mtb, *M. kansasii*, and *M. bovis*, and 1 h for *M. smegmatis*. For THP-1 macrophages, the same concentration of resazurin was added into the wells, and fluorescence signal was recorded 2 h after incubation.

### Mtb viability assays

#### Luminescence

The direct killing ability of the compounds was assessed by the luminescence signal readout of Mtb-*lux*. Mid-log Mtb-*lux* was diluted to an OD of 0.03 and was added into 96-well white plates at 100 µL in each well. One hundred microliters of serially diluted BPD-6 and 9 (12.5–200 µM), RIF (1.2–2.4 µM, Fisher Scientific), and INH (3.6 µM, Acros Organics) were added into the wells. A readout of luminescence signal was performed

immediately after plate setup (time 0). The plates were incubated in 37°C without shaking for 20 h, followed by a final readout. Luminescence was measured with the Synergy H1 Microplate Reader with an optimized integration time of 10 s.

### Colony forming unit (CFU) plating

Middlebrook 7H10 agar plates (BD Biosciences) supplemented with 0.5% glycerol, 10% OADC, 24 µg/mL D-pantothenic acid, and 50 µg/mL L-leucine were prepared. Mid-log Mtb was washed and treated with serially diluted compounds (50–200 µM), RIF (0.3–0.6 µM), and untreated control in 96-well plates. The plates were incubated at 37°C for 20 h. The bacteria in each well were serially diluted by 10-fold. The last four dilutions ($10^{-2}$, $10^{-3}$, $10^{-4}$, $10^{-5}$) were inoculated on 7H10 agar plates. After 3 wk of incubation at 37°C, each plate was counted for colonies and calculated for correlative CFU/mL.

### Intracellular Mtb survival assay

THP-1 cells were differentiated into macrophages as described above. The amount of Mtb-*lux* needed for a multiplicity of infection (MOI) of 10 was prepared using the conversion of $3 \times 10^8$ bacteria/mL for OD 1.0. Log-phase bacteria were resuspended in RPMI 1640 infection media: supplemented with 10% human serum (Millipore Sigma, Burlington, MA, USA), 10 mM HEPES, 2 mM glutamine, D-pantothenic acid (24 µg/mL), and L-leucine (50 µg/mL). After differentiation, the THP-1-derived macrophages were washed with phosphate-buffered saline (PBS) and recovered in infection media for 1 h, and then infected with Mtb-*lux* for 4 h. Extracellular Mtb-*lux* was removed by three PBS washes, followed by addition of compounds. Intracellular survival of Mtb-*lux* was quantified 24 h post-infection by luminescence readouts with the Synergy H1 Microplate Reader with an integration time of 10 s.

### Generation of non-replicating Mtb

Non-replicating Mtb (NR-Mtb-*lux*) was generated using an established low-pH model (7). Mid-log Mtb-*lux* was washed with PBS (1×) and transferred into phosphate-citrate buffer (PCB) media: 1× PCB (composed of 0.2 M sodium phosphate [Fisher Scientific] with 0.1 M citric acid [Fisher Scientific] at pH 4.5) supplemented with 0.05% tyloxapol. The OD was adjusted to 0.5. Bacteria were maintained at 37°C for 7 d. OD was measured every day to confirm the non-growing state. For treatments with compounds, non-replicating Mtb-*lux* was removed from PCB buffer on day 7 and was resuspended in complete 7H9 media directly before compound treatments. RLU viability readouts were measured as described above.

### Measurement of BPD accumulation within bacteria

Log-phase bacteria were washed and resuspended with PBS to OD 0.1 in Eppendorf tubes, and then treated with the compounds at the indicated concentrations at 37°C. At various time points, the bacteria-compound suspension was centrifuged at 8,000 × $g$ for 3 min. Supernatants were transferred into 96-well plates, while the remaining pellets were washed two times with PBS. After the last PBS wash, the pellets were resuspended and transferred into 96-well plates. Fluorescence was measured at an excitation/emission of 420/485 nm using the Synergy H1 Microplate Reader. Mtb-*lux* were treated with 50 µg/mL gentamicin (Calbiochem, San Diego, CA, USA) for 24 h to generate killed Mtb. The RLU of the killed Mtb was measured to confirm the death of the bacteria.

### Fluorescence microscopy

Mtb-RFP were treated with 5 µM BPD-6, BPD-9, or solvent for 4 h. Bacteria were washed with PBS (1×, supplemented with 0.05% Tween 80) after incubation and were fixed with 4% formaldehyde for 15 min. Slides were dried at 37°C and mounted with FluoroSave Reagent (Calbiochem). Samples were imaged with Invitrogen EVOS FL Auto Imaging

System equipped with 100× objective, GFP filter (excitation/emission: 470/525 nm), and RFP filter (excitation/emission: 530/593 nm). Images were analyzed with ImageJ software.

### Generation of BPD-9-resistant strains

Mtb H37Rv was passaged in 7H9 media in the presence of BPD-9 at increasing concentration (2× MIC to 10× MIC) for a total of two or three passages. Bacterial growth was monitored during the passages with the REMA. Cultures were then streaked on 7H10 plates without antibiotic. Isolated colonies were then cultured in media with 25 µM BPD-9 for 7 d. Cultures that maintained growth were considered potential resistant clones and were sent for whole genome sequencing.

### Whole genome sequencing of resistant clones

Genomic DNA (gDNA) was extracted using the Qiagen QIAamp UCP Pathogen Mini kit (Qiagen) with a modified mechanical lysis protocol as previously described (55). Whole genome sequencing was performed using Illumina technology (NovaSeq 6000 instrument) with sequencing libraries prepared using the NEBNext Ultra II DNA Library Prep Kit (NEB). All raw reads (paired end) were analyzed using tools in the Galaxy workflow platform (https://usegalaxy.org/) using the default parameters. First, raw reads were adapter- and quality-trimmed with Trimmomatic v0.38.1 and were mapped onto the Mtb H37Rv reference genome (RefSeq NC_000962.3) using Snippy v4.6.0 for variant calling. The vcf files were then filtered using the TB Variant Filter v0.4.0. All putative variants unique to the mutant strains were manually checked by inspecting the alignments.

### Generation of rv3066 and mmr knockout strains

The Δ*rv3066* and Δ*mmr-rv3066* strains were generated using oligonucleotide-mediated recombineering followed by Bxb1 integrase targeting system (ORBIT) (56). First, Mtb H37Rv was electroporated with 200 ng of the plasmid pKM444. Bacteria were recovered in 7H9 media overnight at 37°C and then plated on 7H10 agar containing 50 µg/mL kanamycin. Plates were incubated at 37°C for 4 wk. Colonies were then picked, and the presence of the pKM444 plasmid was confirmed in the H37Rv-pKM444 clones by PCR by amplification of the kanamycin resistance cassette using the primers kanR-F and kanR-R (Table S2). To obtain the strains Δ*rv3066* and Δ*mmr-rv3066*, recombineering was performed as previously described for Mtb with minor modifications (56). Briefly, 1 µg of *attP*-containing oligonucleotides (Table S2) and 200 ng of pKM496 (zeocin resistance) were co-electroporated in an overnight anhydrotetracycline-induced culture of H37Rv-pKM444 grown at 37°C. Bacteria were then shaken at 37°C in 5 mL 7H9 complete media for 3 d. Recovered bacteria were plated on 7H10 agar plates containing 50 µg/mL zeocin for 4 wk at 37°C. Recombinant candidate colonies were picked, and integration of the pKM496 plasmid was confirmed by PCR using oligonucleotides 300–500 bp upstream and downstream of the integration site (Table S2).

### qPCR

Cells were harvested by centrifugation, and pellets were stored in 1 mL of TRIzol reagent (Thermo Fisher Scientific) at −80°C until further processing. The cells were lysed by bead-beating, and total RNA was extracted by phenol-chloroform with TURBO DNAse treatment (Invitrogen). cDNA was prepared from total RNA using a SuperScript III First-strand Synthesis kit (Invitrogen) and was analyzed by qPCR, in duplicate, for *mmR* expression using Power SYBR GreenPCR Master Mix (Applied Biosystems) on a 7500 Fast Real-Time PCR System (Applied Biosystems). The housekeeping gene *sigA* was used for normalization, and the ΔΔCt method was used for quantification.

## In vivo toxicity tests

Female BALB/c mice of 8–10 wk (Charles River Laboratories) were randomized and treated with different doses of BPD-9 up to 10 mg/kg via intraperitoneal injection ($n$ = 3 or 4). As vehicle, PBS was used. The treatments were administered daily or every second day (depending on the experiment), for a total of 14 d, alternating the injection sites (right and left abdomen). Body weight measurement and visual examination were conducted daily to assess the health and well-being of the mice and to detect any sign of toxicity. The mice were euthanized 24 h after the last treatment, and necropsy was performed to examine the organs.

## In vivo CFU analyses

Female BALB/c mice of 8–10 wk (Charles River Laboratories) were infected via tail vein injection (i.v.) with ~$10^7$ CFU of *M. bovis* BCG, previously resuspended in PBS with 0.05% Tween 80. The animals were subsequently randomized ($n$ = 5) to receive vehicle (PBS) or BPD-9 at 5 mg/kg. The treatments were administered via daily intraperitoneal injection, for a total of 8 d, alternating the injection sites. The animals were monitored daily, recording their body weight. A control cohort ($n$ = 3) was euthanized at 1 d post-infection to ensure a uniform initial infection rate. All the other mice were euthanized on day 8 post-infection. The spleens and lungs were collected and homogenized in 0.05% Tween 80 in PBS and were serially diluted for CFU plating. For *in vivo* CFU analyses, Middlebrook 7H10 agar plates (BD Biosciences) were further supplemented with a mix of antibiotic and antimitotic agents, including polymyxin B (40,000 U/L), amphotericin B (5 mg/L), nalidixic acid (5 mg/L), and azlocillin (5 mg/L), to prevent contamination.

## MIC and IC$_{50}$ modeling

Bacterial and macrophage viability curves were analyzed using GraphPad Prism Version 10. Nonlinear regression analysis was performed on the data using the "[Agonist] vs response – Find ECanything," and the resulting curve fits were used to determine the MIC$_{50/90}$ or IC$_{50}$ values. When the range of test concentrations was high as in Figure 8, the "Gompertz equation for MIC modeling" equation (57) was used to account for the higher range of concentrations.

## Statistical analysis

Statistical analysis was performed using the unpaired Student's *t*-test or one-way analysis of variance (ANOVA) on GraphPad Prism Version 10. Values of $P < 0.05$ were considered as statistically significant.

### ACKNOWLEDGMENTS

We thank Dr. Thien-Fah Mah and Dr. Subash Sad at the University of Ottawa for the generous gift of bacterial strains used in this study. We also acknowledge and thank the Containment Level 3 Technology Platform of the Research Institute of the McGill University Health Centre (RI-MUHC).

This work was supported by grants from the Canadian Institutes of Health Research (CIHR, PJT-162424 and PPE-185827), the National Sanitarium Association Scholars Program and the SIMM-uOttawa Joint Research Centre on Systems and Personalized Pharmacology (SIMMUO201801, 201902, 202001, 202102) to J.S., the NSFC (82373708, 22171275) and Shanghai Academic/Technology Research Leader (23XD1424400) to W.Y., and a CIHR Foundation Grant (FDN-148362) to M.A.B. YC.L. was supported by the Canada Graduate Scholarships (CIHR CGS-M & NSERC CGS-D) and the Ontario Graduate Scholarship.

## AUTHOR AFFILIATIONS

[1]Department of Microbiology and Immunology, University of British Columbia, Vancouver, Canada

[2]Department of Biochemistry, Microbiology and Immunology, University of Ottawa, Ottawa, Canada

[3]Chinese Academy of Sciences Key Laboratory of Receptor Research, Shanghai Institute of Materia Medica (SIMM), Chinese Academy of Sciences, Shanghai, China

[4]University of Chinese Academy of Sciences, Beijing, China

[5]Infectious Diseases and Immunity in Global Health Program, Research Institute of the McGill University Health Centre, Montréal, Canada

[6]McGill International TB Centre, Montréal, Canada

[7]Department of Microbiology and Immunology, McGill University, Montréal, Canada

[8]Department of Medicine, McGill University Health Centre, Montréal, Canada

[9]School of Pharmaceutical Science and Technology, Hangzhou Institute for Advanced Study, University of Chinese Academy of Sciences, Hangzhou, China

## AUTHOR ORCIDs

Yi Chu Liang  http://orcid.org/0000-0002-0797-126X
Stefania Berton  http://orcid.org/0000-0003-2961-6100
Marcel A. Behr  http://orcid.org/0000-0003-0402-4467
Weibo Yang  http://orcid.org/0000-0003-1633-7655
Jim Sun  http://orcid.org/0000-0002-6873-6052

## FUNDING

| Funder | Grant(s) | Author(s) |
|---|---|---|
| Canadian Government \| Canadian Institutes of Health Research (CIHR) | PJT-162424, PPE-185827 | Jim Sun |
| National Sanitarium Association (NSA) | Scholars Program | Jim Sun |
| Canadian Government \| Canadian Institutes of Health Research (CIHR) | FDN-148362 | Marcel A. Behr |
| STCSM \| Program of Shanghai Academic Research Leader (Shanghai Academic Research Leader) | 23XD1424400 | Weibo Yang |
| MOST \| National Natural Science Foundation of China (NSFC) | 82373708, 22171275 | Weibo Yang |

## AUTHOR CONTRIBUTIONS

Yi Chu Liang, Conceptualization, Funding acquisition, Investigation, Supervision, Validation, Visualization, Writing – original draft, Writing – review and editing | Zhiqi Sun, Conceptualization, Investigation, Visualization, Writing – original draft, Writing – review and editing | Chen Lu, Conceptualization, Investigation, Visualization, Writing – review and editing | Andréanne Lupien, Conceptualization, Investigation, Validation, Visualization, Writing – original draft, Writing – review and editing | Zhongliang Xu, Conceptualization, Investigation, Visualization, Writing – original draft, Writing – review and editing | Stefania Berton, Investigation, Supervision, Visualization, Writing – review and editing | Peng Xu, Conceptualization, Investigation, Visualization, Writing – review and editing | Marcel A. Behr, Conceptualization, Funding acquisition, Project administration, Resources, Supervision, Writing – review and editing | Weibo Yang, Conceptualization, Funding acquisition, Project administration, Resources, Supervision, Visualization, Writing – original draft, Writing – review and editing | Jim Sun, Conceptualization, Funding acquisition, Project administration, Resources, Supervision, Visualization, Writing – original draft, Writing – review and editing

## DATA AVAILABILITY

Sequence data from the BPD-9-resistant mutants have been deposited in the NCBI Sequence Read Archive under the NCBI BioProject ID PRJNA1107169.

## ETHICS APPROVAL

All animal experiments were approved by the Animal Care Committee of the University of Ottawa and were supported by the Animal Care and Veterinary Service under the protocols #3234 and #4216. Housing and experimental procedures were in accordance with the institutional guidelines, the Animals for Research Act, and the Canadian Council on Animal Care.

## ADDITIONAL FILES

The following material is available online.

### Supplemental Material

**Supplemental material (Spectrum01246-24-s0001.docx).** Fig. S1 to S7; Tables S1 and S2.

### Open Peer Review

**PEER REVIEW HISTORY (review-history.pdf).** An accounting of the reviewer comments and feedback.

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
