## [Reviewer comments · Microbiology Spectrum]

Microbiology Spectrum

Discovery of benzo[c]phenanthridine derivatives with potent activity against multidrug resistant *Mycobacterium tuberculosis*

Yi Chu Liang, Zhiqi Sun, Chen Lu, Andréanne Lupien, Zhongliang Xu, Stefania Berton, Peng Xu, Marcel Behr, Weibo Yang, and Jim Sun

Corresponding Author(s): Jim Sun, The University of British Columbia

Review Timeline:

Submission Date:	May 20, 2024
Editorial Decision:	July 2, 2024
Revision Received:	July 23, 2024
Accepted:	August 17, 2024

Editor: Varadharajan Sundaramurthy

Reviewer(s): The reviewers have opted to remain anonymous.

Transaction Report:

DOI: <https://doi.org/10.1128/spectrum.01246-24>

Re: Spectrum01246-24 (Discovery of benzo[c]phenanthridine derivatives with potent activity against multidrug resistant *Mycobacterium tuberculosis*)

Dear Dr. Jim Sun:

Thank you for the privilege of reviewing your work. Below you will find my comments, instructions from the Spectrum editorial office, and the reviewer comments.

Please address the reviewers comments and return the manuscript within 60 days; if you cannot complete the modification within this time period, please contact me. If you do not wish to modify the manuscript and prefer to submit it to another journal, notify me immediately so that the manuscript may be formally withdrawn from consideration by Spectrum.

Revision Guidelines

Sincerely,
Varadharajan Sundaramurthy
Editor
Microbiology Spectrum

Reviewer #1 (Comments for the Author):

Drug resistance is a huge challenge in treatment of TB infections. New molecules and drug targets will help us address some of these challenges. In this study, the authors report on the anti-mycobacterial activity of class of molecules, BPDs. These molecules are derivatives of a previously described anti-bacterial natural product, sanguinarine. Using structural guided chemistry, they synthesized 35 new derivatives around the base structure. 5 of these derivatives were found to exhibit potent and specific activity against Mtb. Two of these derivatives, BPD-6 and 9 were then characterized for their activity in vitro against actively replicating and non-replicating Mtb and for intracellular activity in infected macrophages. Some activity was also

documented in vivo, in lungs of infected mice. Generation of resistant mutants revealed the role of a drug efflux pump in resistance. Overall this is a very well designed and executed study, characterizing the profile of new class of compounds targeting Mtb. The manuscript is well written and the experimental details well described.

Specific comments:

- It is not clear from the data provided why the authors ruled out ftsZ as a putative target of BPD compounds. The natural product sanguinarine has been shown to bind ftsZ. While ftsZ is indeed conserved across bacterial species, compounds can have species specific activity against a particular bacteria.
- Figure 2A. BPD-9 has a unique activity at sub-MIC concentrations compared to BPD-6,29 and 32. It does not exhibit peak at sub-MIC concentration. Would this indicate that BPD-9 has additional activity separate from the ones shared with the other molecules. The authors could provide a discussion on this.
- There was no clear explanation why different fold-MICs were used in different experiments. For example, in Fig. 2B,C, 10X MIC of INH and RIF were used, but then in Fig 2D, 5X MIC is used. Also for consistency, might be useful to provide INH data in Fig. 2D as well.
- In the experiments with the NR-media, Mtb bacteria were exposed to the compounds in complete media. Is this necessary for activity of the compounds? If the compounds were added directly to Mtb bacteria, while still in the NR-media, would the compounds still cause growth inhibition upon recovery into fresh media. This could help disentangle if the compounds are active on NR-bacteria or they prevent the resuscitation of NR-bacteria. At present, therefore, the statement made on Page 10, "Given that the BPD compounds inhibit reactivation of NR-Mtb.....", is speculative.
- Figure 7A. BPD6/9 uptake was visualized by fluorescence microscopy. The authors concluded that "...a portion of the Mtb did not appear to accumulate the compounds...". This is actually not clear from the images shown in Figure 7A. It appears that there is heterogeneity in uptake of the compound. Some individual bacteria take up the compound and some do not as is clearly seen in the middle panel.
- Table 4, lists the mutations identified in the four resistant mutant strains, however only the rv3066 mutation is studied and discussed. It would be useful to at least provide some discussion on the mutations identified in the other four loci shared amongst some of the mutant strains.
- Figure 8. It is interesting that the strain carrying the double mutation of mmr and rv3066 has similar MIC to BPD9 as the wild-type strain suggesting that the MMR efflux pump probably has no activity in wild-type cells. Now that the mmr mutant strain is available, it would have been useful to generate resistant strains against BPD9 in this background to identify the true molecular target of the compound.

Minor comments:

- Page 7. Typo -ftsZ is misspelt as "fstZ"

Reviewer #2 (Comments for the Author):

This study presents impressive research on synthesizing and evaluating 35 novel benzophenanthridine derivatives (BPD) targeting Mtb. The authors identified the two derivatives BPD-6 and BPD-9 as most active and continued characterization of those variants.

These compounds show strong inhibitory activity against various Mycobacterium strains and demonstrate efficacy in reducing Mtb survival within infected macrophages. Notably, BPD-9 also reduced the burden of M. bovis BCG in infected mice, indicating its potential as a therapeutic agent.

While there were some issues with cytotoxicity, the study's in vivo results are promising. Importantly, BPD-6 and BPD-9 exhibited potent activity against multiple multidrug-resistant (MDR) Mtb clinical isolates. Overall, this manuscript offers a significant contribution to TB drug discovery, with clear potential for further development and application.

Major points:

- Summary in Introduction: The summary part in the introduction is too long, detailed, and repetitive. Consider condensing it for clarity.
- Reduced Cytotoxicity: In the introduction and several parts of the manuscript, "reduced cytotoxicity" is mentioned. Specify that this reduction is in comparison to the original SG extracts.
- Page 5 - MIC90 of SG: The MIC90 of SG is stated as 85 μ M. This concentration was not tested in Fig 1A, suggesting it was extrapolated using software. The authors should plot the curve used for this calculation, as done for other graphs.
- Table 1 - MIC90 Inconsistency: Table 1 lists the MIC90 of SG as 74 μ M, differing from the 85 μ M stated earlier. Clarify this inconsistency and ensure consistency throughout the manuscript.
- Figures 2 and 3 - MIC Values: The text mentions that control concentrations of INH and RIF represent fold-MIC90s, but their actual MICs are not provided. Adding fold-MIC90s for all compounds inside of the Figures 2 and 3 would enhance clarity.
- Resistance Development: The statement, "We also speculate that resistance to these compounds could be much slower to develop," is intriguing but needs elaboration. Support this claim with references to relevant literature if possible.
- Figure 3 Legend: The figure legend refers to "latent Mtb," which is inconsistent with the term "NR-Mtb" used elsewhere. Ensure

consistent terminology throughout, and clarify that the conditions describe NR-Mtb, not latent Mtb. Latent TB is reserved for in vivo infection experiments.

- Figure 5 Legend: Refer to *M. bovis* BCG instead of Bacillus Calmette-Guérin or BCG strain for consistency.
- Supplementary Information: Include a table listing all strains used in the study, detailing their genetic backgrounds.

Minor points:

- Compound Efficacy: It is unclear whether the compounds prevent reactivation/resuscitation of NR-Mtb or kill non-growing Mtb. Use the distinct activities of control antibiotics INH and RIF to draw clearer conclusions. INH showed no activity to NR-Mtb suggesting the BPD seem to target NR-Mtb.
- Table 1 Notation: Replace "n.a." with ">100 μ M" for clarity. N.a is must often used for not available.
- Strain Comparisons: When comparing NR-Mtb and Mtb-lux, use consistent terminology. Refer to these as NR-Mtb-lux vs Mtb-lux, or simply Mtb vs NR-Mtb to highlight the growth state focus.
- In Vivo Data: The activities of BPD-6 and BPD-9 are similar. Explain also in the manuscript why only BPD-9 was tested in vivo.
- Negative Synergistic Effects: Refer to these as "antagonistic effects."
- Antibiotic Interaction Studies: These effects are minimal and should be moved to the supplementary section.
- Compound Accumulation: Clarify what "relatively quick" accumulation means by providing a comparison or context, maybe by comparison with other antibiotics if possible.
- Staining and Metabolic State: Not all bacteria stain with the compound, possibly due to different metabolic states. Are the Gentamycin killed bacteria more uniform stained? This could provide additional insights.
- Therapeutic Window: The therapeutic window between BPD-6, BPD-9, and macrophages is solid data, however the window is rather small. Interpret these results carefully rather than emphasizing potential for TB treatment.

Point-by-point response to reviewer comments (MS#: Spectrum01246-24)

Reviewer #1 (Comments for the Author):

Drug resistance is a huge challenge in treatment of TB infections. New molecules and drug targets will help us address some of these challenges. In this study, the authors report on the anti-mycobacterial activity of class of molecules, BPDs. These molecules are derivatives of a previously described anti-bacterial natural product, sanguinarine. Using structural guided chemistry, they synthesized 35 new derivatives around the base structure. 5 of these derivatives were found to exhibit potent and specific activity against Mtb. Two of these derivatives, BPD-6 and 9 were then characterized for their activity in vitro against actively replicating and non-replicating Mtb and for intracellular activity in infected macrophages. Some activity was also documented in vivo, in lungs of infected mice. Generation of resistant mutants revealed the role of a drug efflux pump in resistance. Overall this is a very well designed and executed study, characterizing the profile of new class of compounds targeting Mtb. The manuscript is well written and the experimental details well described.

Specific comments:

1. It is not clear from the data provided why the authors ruled out ftsZ as a putative target of BPD compounds. The natural product sanguinarine has been shown to bind ftsZ. While ftsZ is indeed conserved across bacterial species, compounds can have species specific activity against a particular bacteria.

We agree with the Reviewer that it is premature to rule out ftsZ as a target of BPDs. A protein sequence alignment between FtsZ from Mtb and the other bacteria species (Table 1) shows that Mtb FtsZ has high homology with FtsZ from other mycobacteria (>90% similarity and identity) and much lower homology to FtsZ from other bacterial species (~65-70% similarity and ~48-55% identity). As the Reviewer pointed out, it remains possible that BPD targets mycobacterial FtsZ while remaining ineffective against other bacterial FtsZ due to sequence differences of the protein. We have revised the text to discuss this point on **Page 6 in lines 142-148**.

2. Figure 2A. BPD-9 has a unique activity at sub-MIC concentrations compared to BPD-6,29 and 32. It does not exhibit peak at sub-MIC concentration. Would this indicate that BPD-9 has additional activity separate from the ones shared with the other molecules. The authors could provide a discussion on this.

We believe that it is unlikely BPD-9 has a separate unique activity from the rest of the compounds. Rather, we hypothesize that it is just a little better. The sub-MIC activity is reflected in the lower MIC₅₀ value of BPD-9 compared to BPD-6 (~3-fold lower, Table 2). However, this difference is in the same ballpark as the MIC₉₀ value of BPD-9 (~1.7-fold lower, Table 1). The peaks that appear at lower concentrations are a reflection of the assay variability that could be smoothed out with more concentrations tested in that range. Even for BPD-9, a 'peak' forms at the 1.25 μ M data point, but the compounds activity takes effect at the next concentration.

3. There was no clear explanation why different fold-MICs were used in different experiments. For example, in Fig. 2B,C, 10X MIC of INH and RIF were used, but then in Fig 2D, 5X MIC is used. Also for consistency, might be useful to provide INH data in Fig. 2D as well.

We thank the Reviewer for this feedback. In experiments where we evaluate the inhibition activity of BPD compounds such as Figures 2B-C and 3B-C, we used a concentration of 10X MIC for RIF/INH as a way to highlight the activity of BPD compounds relative to these established antibiotics. We wanted to emphasize

the point that even when used at 1-5X MIC, BPD compounds were comparable to or showed enhanced inhibition compared to even 10X MIC of INH or RIF. In Figure 2D, we aimed to determine if the BPD compounds are bactericidal. For this purpose, it is generally accepted that bactericidal compounds must achieve at least ~2-log reduction in CFU at a concentration within 4X its MIC (MBC₉₀) (1, 2), which is why we chose to use a concentration of RIF near its reported MBC₉₀ (5X MIC). Furthermore, we did not include INH in this bactericidal test because we considered RIF as a better control for a bactericidal antibiotic (3), whereas INH could be bactericidal or bacteriostatic depending on the growth/metabolism of mycobacteria. We have now revised the manuscript to add relevant text to properly explain the choice of fold-MIC for Figures 2 and 3 (**Pages 6-7 in lines 156-158, 164-166, and 186**). In addition, and as per request of Reviewer 2 (Comment #5), we have also annotated **revised Figures 2B-D and 3B-C** with the approx. fold-MIC concentrations used in the experiments to add clarity.

4. In the experiments with the NR-media, Mtb bacteria were exposed to the compounds in complete media. Is this necessary for activity of the compounds? If the compounds were added directly to Mtb bacteria, while still in the NR-media, would the compounds still cause growth inhibition upon recovery into fresh media. This could help disentangle if the compounds are active on NR-bacteria or they prevent the resuscitation of NR-bacteria. At present, therefore, the statement made on Page 10, "Given that the BPD compounds inhibit reactivation of NR-Mtb.....", is speculative.

We thank the Reviewer for raising this important point that also aligns with comment #10 from Reviewer 2. We agree with both Reviewers that the experiment shown in Figure 3 cannot distinguish between whether the compounds are active on NR-Mtb or prevent resuscitation of NR-Mtb. We did not treat the bacteria in the NR-media since there would not be sufficient window to measure decreases in metabolic activity (at the detection limit of Mtb-lux), which would have necessitated a CFU plating readout. Additionally, as the Reviewer pointed out, the activity of the compounds in NR-media could also be altered, particularly due to the low pH, which has also been reported to inhibit activity of some TB antibiotics like bedaquiline and metronidazole (4). We agree that our data does not prove that BPD compounds can directly kill NR-Mtb, which while possible and enticing, would require further experimentation to validate. To address this point, we have now revised the manuscript to add a discussion on this and to clarify our statement that BPD compounds, as shown in the experiment in Figure 3, prevent the resuscitation of NR-Mtb. The new discussion can be found on **Page 7 in lines 202-204, 207, and 209-210**.

5. Figure 7A. BPD6/9 uptake was visualized by fluorescence microscopy. The authors concluded that "...a portion of the Mtb did not appear to accumulate the compounds...". This is actually not clear from the images shown in Figure 7A. It appears that there is heterogeneity in uptake of the compound. Some individual bacteria take up the compound and some do not as is clearly seen in the middle panel.

We agree with the Reviewer that images in Figure 7A show an uneven distribution of bacteria that accumulate the compound and bacteria that do not accumulate the compound. While this is what we intended to convey with the original statement, we agree the statement as written was confusing. We have revised the text to improve clarity and to describe/discuss this heterogeneity in compound accumulation on **Page 9 in lines 280-282 and Page 10 in lines 301-306**. Given that accumulation of the compound is greater in live compared to killed Mtb, and that only live Mtb treated with BPD compounds result in the major decrease in signal at 24 hours, it suggests that the metabolic state of the bacteria affects a combination of the accumulation, degradation, and/or efflux of BPD within the bacteria. Due to the heterogenous nature of each bacillus in culture, it is possible that the bacteria that did not accumulate

the compound in the images were at a stage of growth with relatively decreased metabolic activity (slower uptake) or at increased metabolic activity (metabolized or degraded the compound, therefore lost signal). Furthermore, it is also possible that there is heterogeneous expression of the *mmr* efflux pump that we later showed to be involved in BPD-9 resistance.

6. Table 4, lists the mutations identified in the four resistant mutant strains, however only the *rv3066* mutation is studied and discussed. It would be useful to at least provide some discussion on the mutations identified in the other four loci shared amongst some of the mutant strains.

As the Reviewer mentioned, we focused on *rv3066*, as mutations in this gene were observed in all 4 BPD-9-resistant mutants sequenced. Also, we showed that the BPD-9 resistance in the resistant mutants (R1- to R4_BPD-9 mutants, MIC₉₀ ranging from 50.2 to 55.2 μM) was fully recapitulated when *rv3066* was deleted (Figure 8B). Upon review of this results section, we realized the MIC₉₀ values for the 4 resistant mutants were missing, so we have added the MIC₉₀ values to **revised Table 4**. This result suggests that mutations in *rv3066* and the resulting Mmr efflux pump overexpression (Figure 8A) is responsible for BDP-9 resistance in the resistant mutants. The other mutations observed may be compensatory mechanisms that do not directly contribute to BPD-9 resistance in these strains. R1_BPD-9 contains a synonymous mutation in *Rv2678*, which encodes for a probable uroporphyrinogen decarboxylase HemE. We identified 3 non-synonymous mutations in *rv1960* (Arg51Gly), *rv2678* (Glu898fs), and *rv2962* (Thr234Ile). Based on transposon mutagenesis analysis, all three genes are non-essential for *in vitro* growth in *Mtb* H37Rv. *rv1960* and *rv2962* encode, respectively, a possible antitoxin ParD1, part of the type II toxin-antitoxin system (5), and a possible glycosyl transferase implicated in intermediary metabolism and respiration. Although the expression of *rv2962* in *M. smegmatis* increased bacterial survival in macrophages (6), the function of these two genes in *Mtb* is still unclear. However, the function of *Rv2933* has been explored and has been shown to encode for a phenolphthiocerol synthesis type-I polyketide synthase PpsC involved in the biosynthesis of phenolphthiocerol and phthiocerol dimycocerosate (PDIM). The Glu898fs mutation observed in R1_ and R3_BDP-9 mutants is next to a region recently described as a 7-cytosine homopolymeric tract and a potential ‘hotspot’ for mutation. This mutation may have been selected in our mutants due to the lack of a DNA mismatch repair in *Mtb* (7). We added a succinct description and discussion of the other mutations observed in the manuscript on **Pages 11 in lines 341-348**.

7. Figure 8. It is interesting that the strain carrying the double mutation of *mmr* and *rv3066* has similar MIC to BPD9 as the wild-type strain suggesting that the MMR efflux pump probably has no activity in wild-type cells. Now that the *mmr* mutant strain is available, it would have been useful to generate resistant strains against BPD9 in this background to identify the true molecular target of the compound.

We agree with the Reviewer that generating resistant mutants against BPD9 in the *mmr* knockout background is a logical next step. Indeed, this strategy is the beginning of a new project that we hope will reveal the mode of action for BPD-9. The lack of increased sensitivity to BPD-9 with the deletion of the *mmr* efflux pump, could be as the Reviewer points out, due to marginal efflux of the compounds by Mmr owing to its low expression in wild-type *Mtb*.

Minor comments:

8. Page 7. Typo -ftsZ is misspelt as "fstZ".

We have corrected the spelling mistake. **(Page 6 in lines 141-143 and 145-147)**

Reviewer #2 (Comments for the Author):

This study presents impressive research on synthesizing and evaluating 35 novel benzophenanthridine derivatives (BPD) targeting Mtb. The authors identified the two derivatives BPD-6 and BPD-9 as most active and continued characterization of those variants. These compounds show strong inhibitory activity against various Mycobacterium strains and demonstrate efficacy in reducing Mtb survival within infected macrophages. Notably, BPD-9 also reduced the burden of M. bovis BCG in infected mice, indicating its potential as a therapeutic agent. While there were some issues with cytotoxicity, the study's in vivo results are promising. Importantly, BPD-6 and BPD-9 exhibited potent activity against multiple multidrug-resistant (MDR) Mtb clinical isolates. Overall, this manuscript offers a significant contribution to TB drug discovery, with clear potential for further development and application.

Major points:

1. Summary in Introduction: The summary part in the introduction is too long, detailed, and repetitive. Consider condensing it for clarity.

We thank the Reviewer for this suggestion and have shortened this section for improved clarity and conciseness (Pages 3-4 in lines 87-97).

2. Reduced Cytotoxicity: In the introduction and several parts of the manuscript, "reduced cytotoxicity" is mentioned. Specify that this reduction is in comparison to the original SG extracts.

Thank you. We have done so in the text. (Page 4 in line 92, and Page 7 in line 212)

3. Page 5 - MIC₉₀ of SG: The MIC₉₀ of SG is stated as 85 μ M. This concentration was not tested in Fig 1A, suggesting it was extrapolated using software. The authors should plot the curve used for this calculation, as done for other graphs.

The Reviewer is correct that we determined MIC values using nonlinear regression analysis. We have added a paragraph on MIC determination in the Materials and Methods section on Page 18 in lines 601-605. We have also revised the Mtb viability curves in Figures 1A, 2A, and 4A with curve fitted versions instead of showing the data as connecting points. This representation is now consistent with the graphs in Figure 8B-C.

4. Table 1 - MIC₉₀ Inconsistency: Table 1 lists the MIC₉₀ of SG as 74 μ M, differing from the 85 μ M stated earlier. Clarify this inconsistency and ensure consistency throughout the manuscript.

We thank the Reviewer for pointing this out this error. We have corrected the value of 85 μ M in the text to 74 μ M, which is the correct value. (Page 5 in line 104)

5. Figures 2 and 3 - MIC Values: The text mentions that control concentrations of INH and RIF represent fold-MIC₉₀s, but their actual MICs are not provided. Adding fold-MIC₉₀s for all compounds inside of the Figures 2 and 3 would enhance clarity.

We agree. We have added the "Approx. fold-MIC₉₀" of the respective compounds used in the figures below the x-axis concentration labels. This has been done for figures that we discuss in the text with references to MIC fold changes of INH and RIF (revised Figures 2B-D and 3B-C).

6. Resistance Development: The statement, "We also speculate that resistance to these compounds could be much slower to develop," is intriguing but needs elaboration. Support this claim with references to relevant literature if possible.

We have removed this statement as we did generate resistance mutants in the later experiments but never compared their rates of development, so our original statement is unsubstantiated.

7. Figure 3 Legend: The figure legend refers to "latent Mtb," which is inconsistent with the term "NR-Mtb" used elsewhere. Ensure consistent terminology throughout, and clarify that the conditions describe NR-Mtb, not latent Mtb. Latent TB is reserved for in vivo infection experiments.

We agree with the Reviewer that consistent terminology should be used throughout the manuscript. We now refer to this strain as "NR-Mtb-*lux*" in the text (Page 7 in lines 183-185, 187, 189, 193, 196, 200, 204-205, 207, and 209, and Page 16 in line 521) and figure legends (Page 23 in lines 794-795 and 797) when we are referring to the specific strain we generated in our study. The term "latent Mtb" has been removed.

8. Figure 5 Legend: Refer to *M. bovis* BCG instead of Bacillus Calmette-Guérin or BCG strain for consistency.

We have revised all instances to "*M. bovis* BCG" as per suggestion by the Reviewer.

9. Supplementary Information: Include a table listing all strains used in the study, detailing their genetic backgrounds.

We thank the Reviewer for this suggestion. We have added this information as **Supplementary Table S1 (Pages S9 and S10)**. The original Supplementary Table S1 (primers list) is now Supplementary Table S2.

Minor points:

10. Compound Efficacy: It is unclear whether the compounds prevent reactivation/resuscitation of NR-Mtb or kill non-growing Mtb. Use the distinct activities of control antibiotics INH and RIF to draw clearer conclusions. INH showed no activity to NR-Mtb suggesting the BPD seem to target NR-Mtb.

We thank the Reviewer for raising this important point and provide below our response to Reviewer 1, Comment #4, that raised the same comment:

We agree with both Reviewers that the experiment shown in Figure 3 cannot distinguish between whether the compounds are active on NR-Mtb or prevent resuscitation of NR-Mtb. We did not treat the bacteria in the NR-media since there would not be sufficient window to measure decreases in metabolic activity (at the detection limit of Mtb-*lux*), which would have necessitated a CFU plating readout. Additionally, as the Reviewer pointed out, the activity of the compounds in NR-media could also be altered, particularly due to the low pH, which has also been reported to inhibit activity of some TB antibiotics like bedaquiline and metronidazole (4). We agree that our data does not prove that BPD compounds can directly kill NR-Mtb, which while possible and enticing, would require further experimentation to validate. To address this point, we have now revised the manuscript to add a discussion on this and to clarify our statement that BPD compounds, as shown in the experiment in Figure 3, prevent the resuscitation of NR-Mtb. The new discussion can be found on **Page 7 in lines 202-204, 207, and 209-210**.

In addition, while we agree with this Reviewer that the lack of activity by INH in our assay supports that BPD may actually target or act on NR-Mtb, we prefer to not over-speculate on this in the absence of data directly showing that BPD acts on NR-Mtb.

11. Table 1 Notation: Replace "n.a." with ">100 μ M" for clarity. N.a is most often used for not available.

As suggested, the notation "n.a" in Table 1 have been replaced (Page 6 located below line 148). As we made this change, we realized that we had previously reported an incorrect value. The compounds were tested up to a concentration of 40 μ M against the non-mycobacterial strains and not 100 μ M, so we corrected this.

12. Strain Comparisons: When comparing NR-Mtb and Mtb-lux, use consistent terminology. Refer to these as NR-Mtb-lux vs Mtb-lux, or simply Mtb vs NR-Mtb to highlight the growth state focus.

We have revised the terminology to call this strain "NR-Mtb-lux" throughout the text and figure legends when referring to our specific strain, which was done on Page 7 (lines 183-185, 187, 189, 193, 196, 200, 204-205, 207, and 209), Page 16 (line 521), and Page 23 (lines 794-795 and 797).

13. In Vivo Data: The activities of BPD-6 and BPD-9 are similar. Explain also in the manuscript why only BPD-9 was tested in vivo.

We agree that both compounds have similar activity in vitro, but BPD-9 possesses a lower MIC and a slightly better therapeutic ratio than BPD-6 (Table 2). We have clarified this in the text (Page 8 in line 237).

14. Negative Synergistic Effects: Refer to these as "antagonistic effects."

We thank the Reviewer for providing the proper terminology and have revised the text accordingly (Page 9 in line 259).

15. Antibiotic Interaction Studies: These effects are minimal and should be moved to the supplementary section.

We appreciate the Reviewer's feedback and agree that the combination effects are low at times, such as in Figure 6A. However, we believe that this data merits being kept as a main figure because the results point out the possibility of BPD compounds in combination therapies, which is an important, if not critical strategy for TB treatment. Importantly, the results also show that combinations of BPD compounds with rifampin do not lead to antagonistic effects. Rifampicin plus amoxicillin or ciprofloxacin have shown antagonism against *Listeria monocytogenes* (8), rifampin and gentamicin have shown antagonism against *Staphylococcus epidermidis* (9). We consider that the minor but statistically significant differences and lack of antagonism in our data (Figure 6) merit publication, but defer to the editor whether this should remain in Figure 6 or be moved to the supplementary data section.

16. Compound Accumulation: Clarify what "relatively quick" accumulation means by providing a comparison or context, maybe by comparison with other antibiotics if possible.

Given that we did not assess the kinetics of the accumulation in comparison to other compounds or antibiotics, we cannot draw conclusions about the speed of uptake. We have revised the sentence to remove references to the speed of accumulation (Page 9 in lines 287-288).

17. Staining and Metabolic State: Not all bacteria stain with the compound, possibly due to different metabolic states. Are the Gentamycin killed bacteria more uniform stained? This could provide additional insights.

We agree with the Reviewer that there is a heterogeneous distribution of the compounds to the Mtb bacteria, which was also observed and commented on by Reviewer 1 (Comment #5). We also agree that differing metabolic states of individual cells within the population could indeed be a contributing factor to this distribution, as we stated in our response to Reviewer 1's comment. The presence of BPD signal

with Mtb is affected by uptake as well as elimination or modification, which are all processes dependent on the metabolic activity of the bacteria. It is possible that bacteria with decreased metabolism have slower uptake, and it is possible that cells with increased metabolism have faster elimination or are at a later stage of compound modification. Both possibilities would lead to different cells presenting less BPD signal than others at a given time. We have expanded our discussion in the text on Page 9 (lines 280-282) and Page 10 (lines 301-306) with these points.

18. Therapeutic Window: The therapeutic window between BPD-6, BPD-9, and macrophages is solid data, however the window is rather small. Interpret these results carefully rather than emphasizing potential for TB treatment.

We agree that the therapeutic ratio of these compounds, while much improved compared to SG, is still relatively small. As such, we have added a statement on Page 8 in lines 224-225 indicating that this is a limitation that needs to be addressed for the translational potential of these compounds.

References

1. Pankey GA, Sabath LD. 2004. Clinical relevance of bacteriostatic versus bactericidal mechanisms of action in the treatment of Gram-positive bacterial infections. *Clinical Infectious Diseases: An Official Publication of the Infectious Diseases Society of America* 38:864-870.
2. Gao W, Kim J-Y, Anderson JR, Akopian T, Hong S, Jin Y-Y, Kandror O, Kim J-W, Lee I-A, Lee S-Y, McAlpine JB, Mulugeta S, Sunoqrot S, Wang Y, Yang S-H, Yoon T-M, Goldberg AL, Pauli GF, Suh J-W, Franzblau SG, Cho S. 2015. The Cyclic Peptide Ecumicin Targeting ClpC1 Is Active against *Mycobacterium tuberculosis* In Vivo. *Antimicrobial Agents and Chemotherapy* 59:880-889.
3. Kaur P, Agarwal S, Datta S. 2009. Delineating Bacteriostatic and Bactericidal Targets in *Mycobacteria* Using IPTG Inducible Antisense Expression. *PLoS ONE* 4:e5923.
4. Early JV, Mullen S, Parish T. 2019. A rapid, low pH, nutrient stress, assay to determine the bactericidal activity of compounds against non-replicating *Mycobacterium tuberculosis*. *PloS One* 14:e0222970.
5. Ziemski M, Leodolter J, Taylor G, Kerschenmeyer A, Weber-Ban E. 2021. Genome-wide interaction screen for *Mycobacterium tuberculosis* ClpCP protease reveals toxin-antitoxin systems as a major substrate class. *The FEBS journal* 288:111-126.
6. Miller BH, Shinnick TM. 2000. Evaluation of *Mycobacterium tuberculosis* Genes Involved in Resistance to Killing by Human Macrophages. *Infection and Immunity* 68:387-390.
7. Mulholland CV, Wiggins TJ, Cui J, Vilchèze C, Rajagopalan S, Shultis MW, Reyes-Fernández EZ, Jacobs WR, Berney M. 2024. Propionate prevents loss of the PDIM virulence lipid in *Mycobacterium tuberculosis*. *Nature Microbiology* 9:1607-1618.
8. Boisivon A, Guiomar C, Carbon C. 1990. In vitro bactericidal activity of amoxicillin, gentamicin, rifampicin, ciprofloxacin and trimethoprim-sulfamethoxazole alone or in combination against *Listeria monocytogenes*. *European Journal of Clinical Microbiology & Infectious Diseases: Official Publication of the European Society of Clinical Microbiology* 9:206-209.
9. Gagnon RF, Richards GK, Kostiner GB. 1994. Time-kill efficacy of antibiotics in combination with rifampin against *Staphylococcus epidermidis* biofilms. *Advances in Peritoneal Dialysis Conference on Peritoneal Dialysis* 10:189-192.

Re: Spectrum01246-24R1 (Discovery of benzo[c]phenanthridine derivatives with potent activity against multidrug resistant *Mycobacterium tuberculosis*)

Dear Dr. Jim Sun:

Your manuscript has been accepted, and I am forwarding it to the ASM production staff for publication. Your paper will first be checked to make sure all elements meet the technical requirements. ASM staff will contact you if anything needs to be revised before copyediting and production can begin. Otherwise, you will be notified when your proofs are ready to be viewed.

Sincerely,
Varadharajan Sundaramurthy
Editor
Microbiology Spectrum

Reviewer #1 (Comments for the Author):

The authors have addressed and suitably responded to most of the comments from the earlier review.

Reviewer #2 (Comments for the Author):

I appreciate the author's attention to detail as they have adequately addressed all of my points. The revised manuscript is now improved in clarity, and presentation, and key discussions are now included.